# ADAPTIVE FLOW MATCHING FOR RESOLVING SMALL-SCALE PHYSICS

## ABSTRACT

Conditioning diffusion and flow models have proven effective for super-resolving small-scale details in natural images. However, in physical sciences such as weather, super-resolving small-scale details poses significant challenges due to: $(i)$ misalignment between input and output distributions (i.e., solutions to distinct partial differential equations (PDEs) follow different trajectories), $(ii)$ multi-scale dynamics, deterministic dynamics at large scales vs. stochastic at small scales, and $(iii)$ limited data, increasing the risk of overfitting. To address these challenges, we propose encoding the inputs to a *latent* base distribution that is closer to the target distribution, followed by flow matching to generate small-scale physics. The encoder captures the deterministic components, while flow matching adds stochastic small-scale details. To account for uncertainty in the deterministic part, we inject noise into the encoder's output using an adaptive noise scaling mechanism, which is dynamically adjusted based on maximum-likelihood estimates of the encoder's predictions. We conduct extensive experiments on both the real-world CWA weather dataset and the PDE-based Kolmogorov dataset, with the CWA task involving super-resolving the weather variables for the region of Taiwan from 25 km to 2 km scales. Our results show that the proposed Adaptive Flow Matching (AFM) framework significantly outperforms existing methods such as conditional diffusion and flows.

## 1 INTRODUCTION

Resolving small-scale physics is crucial in many scientific applications (Wilby et al., 1998; Rampal et al., 2022; 2024). For instance, in the atmospheric sciences, accurately capturing small-scale dynamics is essential for local planning and disaster mitigation. The success of *conditional* diffusion models in super-resolving natural images and videos (Song et al., 2021; Batzolis et al., 2021; Hoogeboom et al., 2023) has recently been extended to super-resolving small-scale physics (Aich et al., 2024; Ling et al., 2024). However, this task faces significant challenges: (*C1*) Input and target data are often *spatially* misaligned due to differing PDE solutions operating at various resolutions, leading to divergent trajectories. For example the eye of the typhoon is spatially misaligned between the low and high resolution simulations in CWA data due to different dynamic models used in each scale (see fig. 1). Additionally, the input and target variables (channels) often represent different physical quantities, causing further misalignment. (*C2*) the data exhibits multiscale dynamics, where certain large-scale processes are more deterministic (e.g. the propagation of midlatitude storms), while small-scale physics, such as thunderstorms, are highly stochastic; and (*C3*) the record of Earth observations is somewhat limited compared to the natural image datasets.

Few efforts have been made to directly address these challenges in generative learning. Prior work typically relies on *residual* learning approaches (Mardani et al., 2023; Zhao et al., 2021). For instance, the method proposed in Mardani et al. (2023) introduces a two-stage process where the deterministic component is first learned through regression, followed by applying diffusion on the residuals to capture the small-scale physics. While this approach offers a way to separate deterministic and stochastic components, it poses a significant risk of overfitting. The initial regression stage may overfit the training data, leading to poor generalization, especially as the data is limited, and thus fails to adequately represent the variability of the small-scale dynamics when training the diffusion model in the second stage. Additionally, this two-stage method lacks a principled way to handle the uncertainty inherent in both the deterministic and stochastic components.

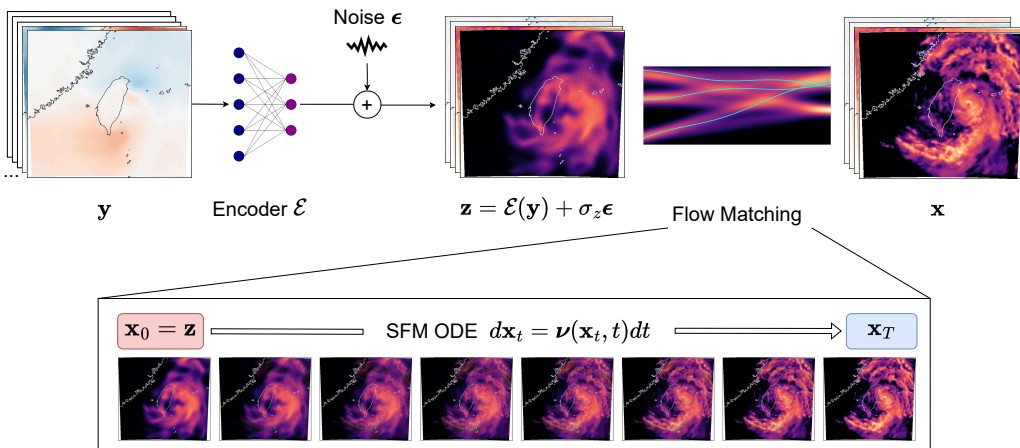

Figure 1: **Overview of the Adaptive Flow Matching (AFM) Method.** The encoder transforms (coarse-res.) inputs into a latent distribution more aligned with the (fine-res.) target. It generates channels absent in the input and corrects both *spatial* and *channel* misalignments, such as repositioning the typhoon's eye to its more accurate location, and generating radar data. From the latent space, FM generates small-scale physics by transporting samples from $p(\mathbf{z})$ to $p(\mathbf{x})$ via the velocity field $\boldsymbol{\nu}(\mathbf{x}, t)$.

To address these limitations, we propose an end-to-end approach based on flow matching. The key elements of our method are as follows: First, an encoder maps the coarse-resolution input data to a *latent* space that is closer to the target fine-resolution distribution. Flow matching is then applied starting from this encoded distribution to generate the target distribution. The encoder captures the deterministic component, which is then augmented with noise to introduce uncertainty. The deterministic prediction is based on the idea that physical processes occur on different time scales, with larger-scale physics having longer-term, more deterministic effects. We refer to this method as Adaptive Flow Matching (AFM), where the stochasticity is controlled by the noise injected at the encoder output. Proper tuning of the noise scale is critical to balance deterministic and stochastic dynamics. To achieve this, we employ a maximum likelihood procedure that adjusts the noise scale based on the encoder's error, dynamically tuning it *on the fly* according to the encoder mismatch.

AFM can be viewed through the lens of diffusion models, efficiently implemented using a stable denoising objective. We conduct extensive experiments on both idealized and realistic datasets. For the realistic data, we use the same data as Mardani et al. (2023), the best estimates of the 25km and 2km observed atmospheric state available from meteorological agencies, centered around a region containing Taiwan. Additionally, we synthesize dynamics from a multi-resolution variant of 2D fluid-flow, where we can control the degree of misalignment. Our results show that AFM consistently outperforms existing methods across various skill metrics.

All in all, the main contributions of this paper are summarized as follows:

- **Adaptive Flow Matching (AFM)**: A method for matching *spatially* misaligned data (plus misaligned *channels*) with multiscale physics, specifically tailored for data-limited regimes in physical sciences.
- **Adaptive Noise Scaling**: We design an adaptive noise scaling based on the maximum likelihood criterion, which optimally balances the learning of deterministic and stochastic components between the encoder and flow matching.
- **Extensive Experiments**: We conduct extensive experiments on multi-scale weather data products as well as synthetic PDE datasets. Our results show that as the degree of misalignment increases, conditional diffusion and flow models perform progressively worse, while our end-to-end AFM framework with adaptive noise avoids overfitting.

## 2   RELATED WORK

**Conditional diffusion and flow models**. Conditioning is a powerful technique for paired image-to-image translation in diffusion and flow models (Batzolis et al., 2021; Kawar et al., 2022; Xingjian &

| Scheme | Perturbation Kernel | Score | Train Loss (Denoising) |
|--------|--------------------|-------|-----------------------|
| CFM | $\mathbf{x}_t = (1-t)\boldsymbol{\epsilon} + t\mathbf{x}$ | $\nabla_{\mathbf{x}_t} \log p_t(\mathbf{x}_t \vert \mathbf{y})$ | $\mathbb{E}\big[\|\mathcal{D}_{\boldsymbol{\theta}}(\mathbf{x}_t; t) - \mathbf{x}\|^2\big]$ |
| CDM | $\mathbf{x}_t = \mathbf{x} + \sigma_t \boldsymbol{\epsilon}$ | $\nabla_{\mathbf{x}_t} \log p_t(\mathbf{x}_t \vert \mathbf{y})$ | $\mathbb{E}\big[\|\mathcal{D}_{\boldsymbol{\theta}}(\mathbf{x}_t; \sigma_t) - \mathbf{x}\|^2\big]$ |
| CorrDiff | $\mathbf{r}_t = \mathbf{r} + \sigma_t \boldsymbol{\epsilon}$ | $\nabla_{\mathbf{r}_t} \log p_t(\mathbf{r}_t \vert \mathbf{y})$ | $\mathbb{E}\big[\|\boldsymbol{e}\|^2\big] \rightarrow \mathbb{E}\big[\|\mathcal{D}_{\boldsymbol{\theta}}(\mathbf{r}_t; \sigma_t) - \mathbf{r}\|^2\big]$ |
| **AFM (Ours)** | $\mathbf{x}_t = \mathbf{x} + \sigma_t(\boldsymbol{e} + \boldsymbol{\epsilon})$ | $\nabla_{\mathbf{x}_t} p_t(\mathbf{x}_t)$ | $\mathbb{E}\big[(\sigma_z/\sigma_t)^2\|\mathcal{D}_{\boldsymbol{\theta}}(\mathbf{x}_t; \sigma_t) - \mathbf{x}\|^2 + \lambda\|\boldsymbol{e}\|^2\big]$ |

Table 1: Comparison between AFM and alternatives for learning the generative map between *misaligned* data $(\mathbf{y}, \mathbf{x})$. Define $\boldsymbol{\epsilon} \sim \mathcal{N}(0, 1)$, $\boldsymbol{e} := (\mathcal{E}(\mathbf{y}) - \mathbf{x})/\sigma_z$, and $\mathbf{r} := \mathbf{x} - \mathbb{E}[\mathbf{x}\vert\mathbf{y}]$. The noise scale $\sigma_z$ is the ML noise estimate ensuring $\mathbb{E}[\|\boldsymbol{e}\|^2] = 1$. CDM and CFM represent conditional diffusion and flow models, respectively.

Xie, 2023). It is commonly used in image restoration tasks such as super-resolution and deblurring, where the goal is to map a low-quality input to a high-quality target, with corresponding pixel associations between input and target. However, as our experiments demonstrate (see section 5), plain conditional models often yield suboptimal performance when data is severely misaligned.

**Diffusion Bridges and Stochastic Interpolants**. Diffusion bridges (De Bortoli et al., 2024; Shi et al., 2023; Liu et al., 2023; Pooladian et al., 2023) facilitate transitions between distributions but rely on assumptions, such as local alignment in I²SB (Liu et al., 2023), unsuitable for *misaligned* data. Stochastic interpolants (Albergo et al., 2023; Albergo & Vanden-Eijnden, 2023; Lipman et al., 2022; Liu et al., 2022) assume smooth transport and are often trained with independent coupling between noise and data, even for the conditional problems. Notably, Albergo et al. (2023) couples base and target distributions. Chen et al. (2024) applies stochastic interpolants to probabilistic forecast of fluid flow. In contrast, our AFM method targets scenarios with misalignment between input and output channels. Specifically, the encoder learns the base distribution from the target's large-scale deterministic dynamics, while adaptively balancing the contributions of deterministic and stochastic components.

**Co-training generative models with encoders**. Co-training encoders with diffusion models has been explored in various domains. DiffCast (Yu et al., 2024) employs an encoder to predict the mean of future frames in precipitation nowcasting, with a diffusion model handling the residuals. Similarly, Grad-TTS (Popov et al., 2021) and Bridge-TTS (Chen et al., 2022) integrate encoders with diffusion-based processes for text-to-speech synthesis. These methods focus on temporal generation or conditional synthesis and operate on a single "channel". In contrast, our AFM framework tackles $(i)$ superresolution and channel reconstruction, $(ii)$ spatial misalignment across different scales, $(iii)$ multiple channels with different stochasticity characteristics.

**Atmospheric super-resolution (downscaling)**. In atmospheric sciences, the task of going from coarse to fine resolution is known as *downscaling*. Several works have explored statistical downscaling using machine learning techniques (see, e.g., Rampal et al. (2024); Wilby et al. (1998); Rampal et al. (2022)). One state-of-the-art method is CorrDiff Mardani et al. (2023), which applies a diffusion model to the residuals left from a deterministic prediction to achieve multivariate super-resolution and new channel synthesis. However, as previously noted, this two-stage approach is prone to severe overfitting, especially in data-limited regimes. To alleviate this issue, CorrDiff uses early stopping when learning the deterministic prediction. In AFM this is achieved using the balance between the deterministic and stochastic errors and the adaptive noise scaling per channel.

## 3 BACKGROUND AND PROBLEM STATEMENT

Consider the task of learning the conditional distribution $p(\mathbf{x}\vert\mathbf{y})$ from a finite set of paired data $\{(\mathbf{y}_i, \mathbf{x}_i)\}_{i=1}^N$, where $\mathbf{y} \in \mathbb{R}^{c \times h \times w}$ and $\mathbf{x} \in \mathbb{R}^{C \times H \times W}$ represent the input and target, respectively. For example, in atmospheric sciences, $\mathbf{y}$ could be a coarse-resolution forecast from the Global Forecast System (GFS) at 25 km resolution, while $\mathbf{x}$ is the fine-resolution target from a high resolution regional weather simulation system at 2 km.

This task is particularly challenging because the pairs $(\mathbf{y}, \mathbf{x})$ are often misaligned. First, these pairs might represent solutions to partial differential equations (PDEs) at significantly different spatial and temporal discretizations. This can lead to completely different temporal or spatial trajectories, including due to effects of internal chaotic dynamics. Second, the input and target may involve

different channels, e.g., corresponding to distinct atmospheric variables, further complicating the learning process.

Before diving into the details of our proposed solution, it is helpful to briefly review flow matching and elucidated diffusion models, which are critical components of our approach.

**Flow Matching (FM)**: Flow matching learns the transformation between two probability distributions by modeling a velocity field $\boldsymbol{\nu}(\mathbf{x}_t, t)$ that transports samples from a source distribution $p_0(\mathbf{x})$ to a target distribution $p_1(\mathbf{x})$. In practice, flows are often trained using a linear interpolant between noise and data. The forward process is described by the ODE:

$$\frac{d\mathbf{x}_t}{dt} = \boldsymbol{\nu}(\mathbf{x}_t, t), \tag{1}$$

where $\boldsymbol{\nu}(\mathbf{x}_t, t)$ represents the velocity field over time $t \in [0, 1]$. For the linear interpolant, the true velocity that generates a single data sample $\mathbf{x}_1$ is given by $\boldsymbol{\nu}_{\text{true}}(\mathbf{x}_t, t) = \mathbf{x}_1 - \mathbf{x}_0$. The goal is to minimize the discrepancy between the learned velocity field $\boldsymbol{\nu}_\theta(\mathbf{x}_t, t)$ and this true velocity per sample:

$$\min_\theta \mathbb{E}_{t, \mathbf{x}_t} \left[ \left\| \boldsymbol{\nu}_\theta(\mathbf{x}_t, t) - (\mathbf{x}_1 - \mathbf{x}_0) \right\|^2 \right]. \tag{2}$$

where $\mathbf{x}_t := (1 - t)\mathbf{x}_0 + t\mathbf{x}_1$. Upon convergence, the learned velocity is used to generate samples by sampling $\mathbf{x}_0 \sim p_0(\mathbf{x})$ and solving the ODE in Eq. 1.

**Diffusion Models (DM)**. Diffusion models generate data by transforming a base distribution, often Gaussian noise, into the target data distribution $p_0(\mathbf{x})$. In the forward diffusion process, Gaussian noise with standard deviation $\sigma$ is added to the data, producing a sequence of distributions $p_0(\mathbf{x}; \sigma)$. As $\sigma$ increases, the data distribution approaches pure noise. The backward process then denoises samples, starting from noise drawn from $\mathcal{N}(0, \sigma_{\text{max}}^2 \mathbf{I})$ and progressively reducing the noise to recover the data distribution.

Considering the variance-exploding elucidated diffusion model (EDM), both the forward and backward processes are described by SDEs. The forward SDE is:

$$d\mathbf{x}_t = \sqrt{2\dot{\sigma}_t \sigma_t} d\boldsymbol{\omega}_t, \tag{3}$$

while the backward SDE is:

$$d\mathbf{x}_t = -2\dot{\sigma}_t \sigma_t \nabla_{\mathbf{x}_t} \log p(\mathbf{x}_t; \sigma_t) dt + \sqrt{2\dot{\sigma}_t \sigma_t} d\boldsymbol{\omega}_t. \tag{4}$$

In EDM, denoising score matching is used to learn the score function $\nabla_{\mathbf{x}} \log p(\mathbf{x}; \sigma)$, essential for the reverse diffusion process. A denoising neural network $D_\theta(\mathbf{x}; \sigma)$ is trained as:

$$\min_\theta \mathbb{E}_{\mathbf{x} \sim p_0} \mathbb{E}_{\sigma_t \sim p_\sigma} \mathbb{E}_{\mathbf{n} \sim \mathcal{N}(0, \sigma_t^2 \mathbf{I})} \left[ \left\| \mathcal{D}_{\boldsymbol{\theta}}(\mathbf{x} + \mathbf{n}, \sigma_t) - \mathbf{x} \right\|^2 \right]. \tag{5}$$

The score function is constructed later via $\nabla_{\mathbf{x}_t} \log p(\mathbf{x}; \sigma_t) = (\mathcal{D}_{\boldsymbol{\theta}}(\mathbf{x}_t, \sigma_t) - \mathbf{x})/\sigma_t^2$.

**Connection between FM and Diffusion Models**. While both methods aim to transform distributions, flow matching does so using ODEs for deterministic evolution, while EDM leverages SDEs for stochastic denoising. An ODE formulation for diffusion models bridges the two methods, allowing flow matching to benefit from formulations and network parameterizations introduced for diffusion models; see e.g., Karras et al. (2022); Song et al. (2021). For the simple noise schedule $\sigma_t = \sqrt{t}$, the ODE for continuous-time EDM is:

$$\frac{d\mathbf{x}_t}{dt} = \frac{\mathbf{x}_t - \mathcal{D}_{\boldsymbol{\theta}}(\mathbf{x}_t, \sigma_t)}{t}, \tag{6}$$

where the right-hand side acts as the velocity field, linking diffusion dynamics to flow matching. Note that the diffusion process runs backward in time from $t = 1$ to $t = 0$, whereas the flow matching process proceeds in the opposite direction.

## 4 ADAPTIVE FLOW MATCHING FOR CONDITIONAL GENERATION

To learn the conditional distribution $p(\mathbf{x}|\mathbf{y})$, one approach is to use conditional diffusion or flow models. These models have been successful in image-to-image tasks like image restoration or super-resolution, where conditioning provides rich information about the target (Saharia et al., 2022).

However, traditional methods struggle when the input $\mathbf{y}$ and target $\mathbf{x}$ are significantly misaligned (see section 5 for evidence). To address this, we propose a multiscale approach:

**Deterministic Dynamics:** The input $\mathbf{y}$ is encoded into a latent variable $\mathbf{z} = \mathcal{E}(\mathbf{y})$. This encoding serves two purposes: $(i)$ it first matches the large-scale, mainly deterministic dynamics of the input and output, aligning the spatially misaligned large-structures due to diverging trajectories, and $(ii)$ it aligns the *channels* by projecting the input into the same space as the output (since $\mathbf{y}$ and $\mathbf{x}$ represent different weather variables).

**Generative Dynamics:** Flow matching is then used to transform the base distribution $p(\mathbf{z})$ into the target distribution $p(\mathbf{x})$. To account for uncertainty in the encoding phase, we perturb the latent variable with Gaussian noise:

$$\mathbf{z} = \mathcal{E}(\mathbf{y}) + \sigma_z \boldsymbol{\epsilon}, \quad \boldsymbol{\epsilon} \sim \mathcal{N}(0, \mathbf{I}). \tag{7}$$

In the following sections, we will detail the learning process for both the encoder and flow matching.

### 4.1 Training

The objective is to jointly learn the encoder and the flow matching model. To achieve this, we delve into flow matching in the latent space. Specifically, it establishes a linear interpolant defined as $\mathbf{x}_t = (1 - t)\mathbf{z} + t\mathbf{x}$, where $\mathbf{z}$ is the encoded state and $\mathbf{x}$ is the target, for $t \in [0, 1]$. Consequently, based on eq. (2), the flow matching objective is formulated as:

$$\min_{\mathcal{E}, \boldsymbol{\theta}} \mathbb{E}_{t, \mathbf{x}, \mathbf{z} \sim \mathcal{N}(\mathcal{E}(\mathbf{y}), \sigma_z)} \left[ \|\boldsymbol{\nu}_\theta(\mathbf{x}_t, t) - (\mathbf{x} - \mathbf{z})\|^2 \right]. \tag{8}$$

Due to the stochasticity in $\mathbf{z}$ and the stability of the EDM framework for training diffusion models—along with its tuning-free characteristics—it is advantageous to incorporate denoising through EDM when training flow matching. To this end, the first step is to recast the linear interpolant as a Gaussian diffusion process (cf. eq. (7)):

$$\mathbf{x}_t = (1 - t)\mathcal{E}(\mathbf{y}) + t\mathbf{x} + (1 - t)\sigma_z \boldsymbol{\epsilon}, \qquad t \in [0, 1] \tag{9}$$

The following proposition simplifies the task of learning the velocity field as a denoising process.

**Proposition 1**. *For the perturbation model $\mathbf{x}_t = \mathbf{x} + \sigma_t \boldsymbol{e} + \sigma_t \boldsymbol{\epsilon}$, where the noise standard deviation is given by $\sigma_t := (1 - t)\sigma_z$, the residual error by $\boldsymbol{e} := (\mathcal{E}(\mathbf{y}) - \mathbf{x})/\sigma_z$, and the noise $\boldsymbol{\epsilon} \sim \mathcal{N}(0, 1)$, the flow matching for joint training of the encoder and flow reduces to the denoising objective:*

$$\min_{\mathcal{E}, \boldsymbol{\theta}} \mathbb{E}_{\mathbf{x}, \mathbf{y}, \sigma_t \sim \mathcal{U}[0, \sigma_z]} \left[ (\sigma_z / \sigma_t)^2 \left\| \mathcal{D}_{\boldsymbol{\theta}}(\mathbf{x}_t, \sigma_t) - \mathbf{x} \right\|^2 \right]. \tag{10}$$

Intuitively, this denoising objective addresses not only the Gaussian noise typical of diffusion models but also residual errors introduced during the deterministic encoding process. Note that the residual error $\boldsymbol{e}$ conveys the essential information about the input conditioning $\mathbf{y}$ required for generating the target $\mathbf{x}$. Therefore, it is crucial to carefully balance the influence of this error by appropriately selecting $\sigma_z$ and applying regularization to the encoding process. These considerations will be discussed next.

### 4.2 Adaptive Noise Scaling

It is essential to tune the noise parameter $\sigma_z$ based on the data before applying diffusion denoising. Specifically, we consider the latent variable:

$$\mathbf{z} = \mathcal{E}(\mathbf{y}) + \sigma_z \boldsymbol{\epsilon}, \tag{11}$$

where we observe $\mathbf{x}$ and $\mathbf{y}$, and aim to adjust $\sigma_z$ in a maximum likelihood (ML) sense so that $\mathbf{z}$ aligns closely with $\mathbf{x}$. In this context, $\sigma_z$ controls the scale of the noise added to the encoder's output $\mathcal{E}(\mathbf{y})$. By leveraging the ML estimator for $\sigma_z$, we can derive it as the root-mean-square-error (RMSE) of the unnormalized residual error, namely

$$\sigma_z = \sqrt{\mathbb{E}[\|\mathbf{x} - \mathcal{E}(\mathbf{y})\|^2]}. \tag{12}$$

To prevent overfitting, the encoder's RMSE is calculated using a validation set every 10k training steps, and we dynamically update $\sigma_z$ based on these measurements as training progresses. Intuitively, if the deterministic regression model overfits to a small training dataset, the validation RMSE

**Algorithm 1** AFM training

1: **Input:** $\lambda, \{(\mathbf{y}_i, \mathbf{x}_i)\}_{i=1}^N$
2: Initialize $\sigma_z, \boldsymbol{\theta}, \mathcal{E}$
3: **repeat**
4:    Sample $\sigma_t \sim \mathcal{U}[0, \sigma_z]$ and $\boldsymbol{\epsilon} \sim \mathcal{N}(0, \mathbf{I})$
5:    Compute error: $\boldsymbol{e} := (\mathcal{E}(\mathbf{y}) - \mathbf{x})/\sigma_z$
6:    Perturb input: $\mathbf{x}_t = \mathbf{x} + \sigma_t(\boldsymbol{e} + \boldsymbol{\epsilon})$
7:    Take a gradient step on:
8:    $\nabla_{\boldsymbol{\theta}, \mathcal{E}}\left[\left(\dfrac{\sigma_z}{\sigma_t}\right)^2 \|\mathcal{D}_{\boldsymbol{\theta}}(\mathbf{x}_t, \sigma_t) - \mathbf{x}\|^2 + \lambda \|\boldsymbol{e}\|^2\right]$
9: **until** convergence

**Algorithm 2** AFM sampling

1: **Input:** $\mathbf{y}, \Delta t, \sigma_z, \mathcal{D}_{\boldsymbol{\theta}}, \mathcal{E}$
2: Sample noise $\boldsymbol{\epsilon} \sim \mathcal{N}(\mathbf{0}, \mathbf{I})$
3: Form latent $\mathbf{z} = \mathcal{E}(\mathbf{y}) + \sigma_z \boldsymbol{\epsilon}$
4: Initialize $\mathbf{x}_0 = \mathbf{z}$
5: **for** $t = 0 : \Delta t : 1$ **do**
6:    $\sigma_t = (1 - t)\sigma_z$
7:    $\boldsymbol{\nu}_\theta(\mathbf{x}_t, t) = (\mathcal{D}_{\boldsymbol{\theta}}(\mathbf{x}_t, \sigma_t) - \mathbf{x}_t)/(1 - t)$
8:    $\mathbf{x}_{t+\Delta t} = \mathbf{x}_t + \boldsymbol{\nu}_\theta(\mathbf{x}_t, t) \cdot \Delta t$
9: **end for**
10: **return** $\mathbf{x}_1$

will grow and thus, our model will adaptively use a higher noise scale in the output of the encoder. At iteration $k$, one can adaptively select $\sigma_z(k)$ using an exponential moving average (EMA), ensuring that the noise scale is continuously updated to reflect the model's performance over time. The adaptive noise scale at iteration $k$ is defined as:

$$\sigma_z \leftarrow (1 - \beta)\sigma_z + \beta\sigma_z(k), \tag{13}$$

**Remark [Alternative Stochastic Encoders].** One could model uncertainty using alternative stochastic encoders like the VAE approach, which imposes KL regularization on the encoder to predict both $\mu$ and $\sigma$, using $\sigma$ to drive the noise for FM. While valid, we opted for a simpler method that seeks an encoder maximizing the likelihood of the output, providing a closed-form solution for $\sigma$ (see Rybkin et al. (2021)). This simplicity aligns with the intuition that a deterministic predictor can effectively predict the output. Although the VAE-based formulation allows automatic learning of $\sigma$, it requires tuning the KL regularization and can suffer from the prior hole problem.

### 4.3 ENCODER REGULARIZATION

Another effective approach to control the residual error $\boldsymbol{e}$ is through regularization. Specifically, one can impose a regression-based regularization term on the encoder, encouraging the output of the encoder to approximate the target, i.e., $\mathbf{x} \approx \mathcal{E}(\mathbf{y})$, thus minimizing the residual error $\boldsymbol{e}$. In an ideal scenario, this would lead to $\boldsymbol{e} \approx 0$. However, enforcing perfect matching can adversely affect the generalization capability of the model, leading to overfitting. To mitigate this, a *soft* regularization is applied, controlled by a penalty weight $\lambda$. This weight balances the trade-off between reducing the residual error and maintaining the generalization ability of the model. The resulting objective function becomes:

$$\min_{\mathcal{E}, \boldsymbol{\theta}} \mathbb{E}_{\mathbf{x}, \mathbf{y}, \sigma_t \sim \mathcal{U}[0, \sigma_z]}\left[\left(\frac{\sigma_z}{\sigma_t}\right)^2 \|\mathcal{D}_{\boldsymbol{\theta}}(\mathbf{x}_t, \sigma_t) - \mathbf{x}\|^2 + \lambda \|\boldsymbol{e}\|^2\right]. \tag{14}$$

where the encoder and denoiser are trained jointly to minimize the denoising and regression losses.

### 4.4 CONNECTIONS TO RESIDUAL LEARNING

Consider the case where we have a pre-trained encoder $\mathcal{E}$, which has been trained using a supervised regression loss, for instance. In this scenario, starting from the forward diffusion process in eq. (9), if we subtract both sides by the encoder output $\mathcal{E}(\mathbf{y})$ and define the residual error as $\boldsymbol{e}_t := \mathbf{x}_t - \mathcal{E}(\mathbf{y})$, we arrive at a form that closely resembles a standard flow matching forward process with Gaussian noise as the base distribution. This is expressed as:

$$\boldsymbol{e}_t = t\boldsymbol{e} + (1 - t)\boldsymbol{\epsilon}, \qquad t \in [0, 1], \tag{15}$$

where $\boldsymbol{e}$ represents the residual error between the target $\mathbf{x}$ and the encoder output $\mathcal{E}(\mathbf{y})$, and $\boldsymbol{\epsilon} \sim \mathcal{N}(0, 1)$ is the noise.

This simple process facilitates the construction of a backward process, where one can learn the velocity field by minimizing the flow matching loss in eq. (2). This approach closely mirrors the

CorrDiff method proposed in Mardani et al. (2023), which leverages residual learning to train diffusion models. CorrDiff has demonstrated considerable success in capturing small-scale details in generative tasks, particularly where precise reconstruction of fine structures is required.

However, as discussed in section 2, the initial supervised training of the encoder often leads to near-perfect matching between $\mathcal{E}(\mathbf{y})$ and $\mathbf{x}$. While this may seem desirable, it can result in overfitting and poor generalization performance. Therefore, balancing this residual learning approach with appropriate regularization is critical to maintaining the model's ability to generalize effectively to unseen data.

### 4.5 SAMPLING

Once the velocity field $\boldsymbol{\nu}_\theta(\mathbf{x}_t, t) = (\mathcal{D}_{\boldsymbol{\theta}}(\mathbf{x}_t, \sigma_t) - \mathbf{x}_t)/(1-t)$ is learned using algorithm 1, sampling simply requires integrating the flow forward in time based on the ODE formulation in eq. (1). The forward integration process from $t = 0$ to $t = 1$ can be expressed as:

$$\mathbf{x}_1 = \int_{t=0}^{t=1} \boldsymbol{\nu}_\theta(\mathbf{x}_t, t) \, dt \tag{16}$$

In practice, this integration can be approximated using Euler steps, which are detailed in algorithm 2 This algorithm outlines the step-by-step procedure for forward sampling through time.

## 5 EXPERIMENTS

We evaluate the performance of the proposed Adaptive Flow Matching (AFM) model on two datasets: Multiscale Kolmogorov Flow and a regional weather downscaling dataset. The weather dataset includes real-world complexity with meteorological observations from Taiwan's Central Weather Administration (CWA). The Multiscale Kolmogorov flow dataset is an idealized set designed to capture misalignment in the Taiwan data. Both datasets present significant downscaling challenges due to scale misalignment, mixed dynamics, and channel-specific variability.

**Baselines**. We compare our AFM model against several baseline approaches, including deterministic and generative methods:

- **Regression**: A standard convolution network (UNet or $1 \times 1$ conv) model trained to predict high-resolution outputs from low-resolution input data using MSE loss training. This serves as a purely deterministic approach, representing a baseline for direct super-resolution.

- **Conditional Diffusion Model (CDM)**: CDM maps Gaussian noise to the high-resolution space while conditioned on the low-resolution input. CDM does not explicitly model the deterministic component in the data.

- **Conditional Flow Matching (CFM)**: A variant of flow matching that interpolates between a Gaussian sample and a data sample, building deterministic mappings.

- **Corrective Diffusion Models (CorrDiff) (Mardani et al., 2023)**: A UNet-based regression network is first trained on pairs $\{(\mathbf{x}_i, \mathbf{y}_i)\}$ using MSE loss to learn the mean $\mathbb{E}[\mathbf{x}|\mathbf{y}]$. The residual error $\boldsymbol{e} := \mathbf{x} - \mathbb{E}[\mathbf{x}|\mathbf{y}]$ is then used to train the diffusion model on the residuals. Early stopping is used to mitigate the overfitting of the UNet regressor.

Unlike our AFM model, which starts from the low-resolution input and learns the stochastic dynamics in the latent space, both CDM and CFM aim to map Gaussian noise directly to the high-resolution output space while being conditioned on the low-resolution input. Conditioning works by concatenating the low-resolution input with the noise, as described in Batzolis et al. (2021) and Saharia et al. (2022). By including these baselines, we evaluate the strengths of our approach against established deterministic and generative methods

**Evaluation Metrics**. We report performance using standard metrics such as RMSE, Continuous Ranked Probability Score (CRPS), Mean Absolute Error (MAE), and Spread Skill Ratio (SSR). These metrics provide a comprehensive assessment of both the accuracy and uncertainty quantification of the model's predictions. RMSE, CRPS, and MAE measure the estimation error while SSR evaluates the model calibration. To calculate CRPS and SSR we produce 64 ensemble members using different seeds. These metrics are discussed in Appendix A.3.

**Network Architecture, Training, and Sampling**. For diffusion model training and sampling, we use EDM Karras et al. (2022), a continuous-time diffusion model available with a public codebase.

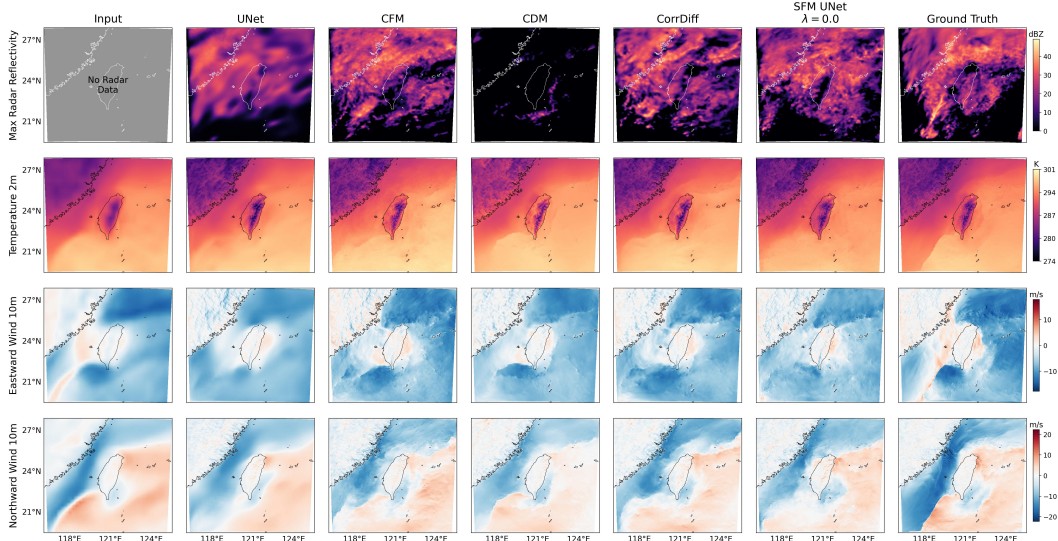

Figure 2: **AFM vs. baselines for different weather variables.** AFM generates more physically consistent outputs, while UNet output appears blurred, and CDM struggles to accurately reconstruct radar reflectivity. Note that radar reflectivity is not present in the input data and is entirely generated as a new channel.

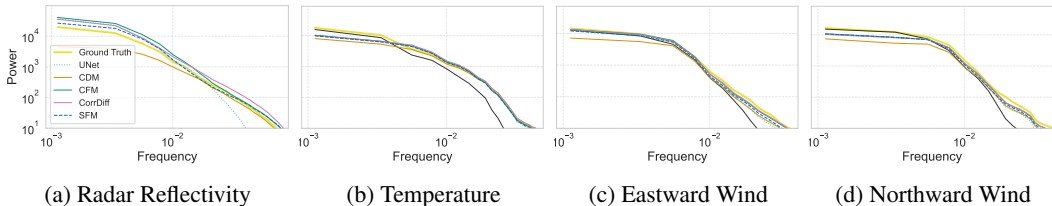

(a) Radar Reflectivity    (b) Temperature    (c) Eastward Wind    (d) Northward Wind

Figure 3: **AFM power spectra vs. baselines for CWA downscaling.** AFM exhibits superior spectral fidelity, closely aligning with the ground truth across all variables, with a particularly strong fidelity for the purely generated radar reflectivity. It consistently outperforms CorrDiff, especially in capturing high-frequency details across all variables.

EDM provides a physics-inspired design based on ODEs, auto-tuned for our scenario (see Table 1 in Karras et al. (2022)). We adopt most of the hyperparameters from EDM and make modifications as detailed in Appendix A.2. Note that EDM is used for CDM and CFM as well in consistent manner.

## 5.1 REGIONAL DOWNSCALING FOR TAIWAN

We focus on the task of super-resolving (downscaling) multiple weather variables for the Taiwan region, a challenging meteorological regime. The input coarse-resolution data at a 25 km scale comes from ERA5 Hersbach et al. (2020), while the target fine-resolution 2 km scale data is sourced from the Central Weather Administration (CWA) Central Weather Administration (CWA) (2021). For evaluation, we use a common set of 205 randomly selected out-of-sample date and time combinations from 2021. Metrics and spectra are computed to compare AFM with baseline models. We utilize a 32-member ensemble; larger ensembles do not significantly alter the key findings. A detailed data description is provided first, followed by our observations.

**Dataset**. The dataset for this study is derived from ERA5 reanalysis data Hersbach et al. (2020), focusing on 12 variables including temperature, wind components, and geopotential height at two pressure levels, as well as surface-level variables like 2-meter temperature and total column water vapor. The target output data Central Weather Administration (CWA) (2021) covers a 900 × 900 km region around Taiwan on a 448 × 448 grid. Hourly observations span four years (2018-2021), split into training (2018-2020) and evaluation (2021) sets. Input data is upsampled using bi-linear interpolation for model consistency Hu et al. (2019); Zhang et al. (2018). See Appendix A.4.1 for the detailed discussion of the datasets.

| Variable | Model | CorrDiff (w/ early stop.) | CFM | CDM | Regressor UNet | AFM (1 × 1 Conv.) |
|---|---|---|---|---|---|---|
| **Radar** | **RMSE** ↓ | 5.08 | 5.06 | 5.70 | 5.09 | **4.90** |
| | **CRPS** ↓ | 1.89 | 1.88 | 2.39 | - | **1.78** |
| | **MAE** ↓ | 2.50 | 2.46 | 2.78 | 2.50 | **2.42** |
| | **SSR** → 1 | 0.38 | 0.33 | **0.46** | - | 0.44 |
| **Temperature** | **RMSE** ↓ | **0.83** | 0.93 | 0.95 | 0.86 | 1.00 |
| | **CRPS** ↓ | **0.50** | 0.58 | 0.54 | - | 0.52 |
| | **MAE** ↓ | **0.60** | 0.72 | 0.70 | 0.62 | 0.67 |
| | **SSR** → 1 | 0.36 | 0.41 | **0.52** | - | 0.47 |
| **Eastward Wind** | **RMSE** ↓ | 1.47 | 1.45 | 1.62 | 1.49 | **1.44** |
| | **CRPS** ↓ | 0.85 | 0.82 | 0.93 | - | **0.80** |
| | **MAE** ↓ | 1.07 | **1.06** | 1.24 | 1.09 | 1.07 |
| | **SSR** → 1 | 0.43 | 0.50 | **0.61** | - | **0.61** |
| **Northward Wind** | **RMSE** ↓ | 1.66 | **1.61** | 1.84 | 1.66 | **1.61** |
| | **CRPS** ↓ | 0.95 | 0.90 | 1.06 | - | **0.88** |
| | **MAE** ↓ | 1.20 | **1.16** | 1.41 | 1.21 | 1.17 |
| | **SSR** → 1 | 0.41 | 0.49 | **0.58** | - | **0.58** |

Table 2: **AFM vs. Baselines for CWA Downscaling:** Values in **bold** show the best performance. AFM outperforms baselines except for the *deterministic* temperature variable, where CorrDiff excels. Temperature, being the most deterministic, benefits from CorrDiff's fully deterministic predictions. While AFM could match CorrDiff using a UNet encoder and higher $\lambda$, this would compromise stochastic predictions for other variables.

### 5.1.1 RESULTS

The performance of AFM is compared with various alternatives, and both deterministic and probabilistic skills are reported in Table 2. We examine three variants of AFM: with ($i$) small versus large encoders; ($ii$) with additional use of adaptive noise scaling; and ($iii$) conditioning in the large encoder limit. Notably, temperature is the most deterministic variable among the four listed, while radar is the most stochastic one.

Our main finding is that AFM consistently outperforms existing alternatives across different metrics for non-deterministic channels (radar and winds). For temperature, although AFM is not the top performer, we can tune $\lambda$ to larger values and achieve performance as good as CorrDiff. This however compromises the stochastic prediction for non-deterministic channels; see the ablation in Table 10 of the Appendix. The ablations are deffered to Appendix B.3 due to space limitations.

Spectral analysis is crucial for assessing fidelity at different scales in weather prediction. Fig. 3 shows that AFM's spectra closely match the ground truth across variables. In contrast, the UNet-based regression scheme fails to generate high-frequency components. Interestingly, the conditional diffusion model (CDM), commonly used for image super-resolution, also lacks spectral fidelity. Regarding the calibration of the generated ensemble (i.e., Spread Skill Ratio, SSR), AFM provides the best balance, being closest to 1.0. While AFM does not eliminate the overall problem of under-dispersive super-resolution, it does improve the balance between the spread and RMSE skill of the generated ensemble, especially for surface wind channels.

### 5.2 MULTISCALE KOLMOGOROV-FLOW

Our Multiscale Kolmogorov Flow dataset offers a simplified simulation of atmospheric dynamics, focusing on downscaling from a coarse to a fine grid while preserving physical structures. Kolmogorov flow (KF) is a well-known scenario where 2D fluid flow in a doubly periodic domain is forced by spatially varying source of momentum. To mimic the structure of the down-scaling problem we couple the KF flow ground truth to an otherwise unforced fluid system representing a coarse-resolution atmospheric simulation. The strength of this coupling $\tau$ controls how well the coarse simulation tracks the ground truth. We are not aware of a similar 2D toy problem for the down-scaling problem, so this setup may be useful for other studies in the area.

**Dataset**. The dataset is constructed by simulating dynamics governed by a system of partial differential equations involving vorticity fields $\zeta_l$ and $\zeta_h$, coupled through parameters like $\tau$ and influenced

| Metric | τ = 3 | | | | τ = 5 | | | | τ = 10 | | | |
|---|---|---|---|---|---|---|---|---|---|---|---|---|
| | **CFM** | **CDM** | **UNet** | **AFM** | **CFM** | **CDM** | **UNet** | **AFM** | **CFM** | **CDM** | **UNet** | **AFM** |
| **RMSE** ↓ | 0.98 | 1.13 | 1.15 | **0.73** | 0.96 | 0.94 | 1.14 | **0.76** | 1.22 | 1.24 | 1.36 | **1.09** |
| **CRPS** ↓ | 0.52 | 0.58 | - | **0.37** | 0.52 | 0.48 | - | **0.40** | 0.67 | **0.65** | - | 0.65 |
| **MAE** ↓ | 0.69 | 0.80 | 0.82 | **0.51** | 0.69 | 0.67 | 0.82 | **0.54** | 0.89 | 0.89 | 1.00 | **0.77** |
| **SSR** → 1 | 0.54 | **0.69** | - | 0.62 | 0.58 | **0.70** | - | 0.58 | 0.56 | **0.76** | - | 0.23 |

Table 3: **AFM vs. baselines for Kolmogorov Flow for various misalignment degrees** τ. AFM consistently demonstrates superior performance across varying levels of data misalignment, showcasing its robustness. While CDM exhibits greater variability, this comes at the expense of significantly reduced fidelity. Note that for deterministic models, CRPS is equivalent to MAE.

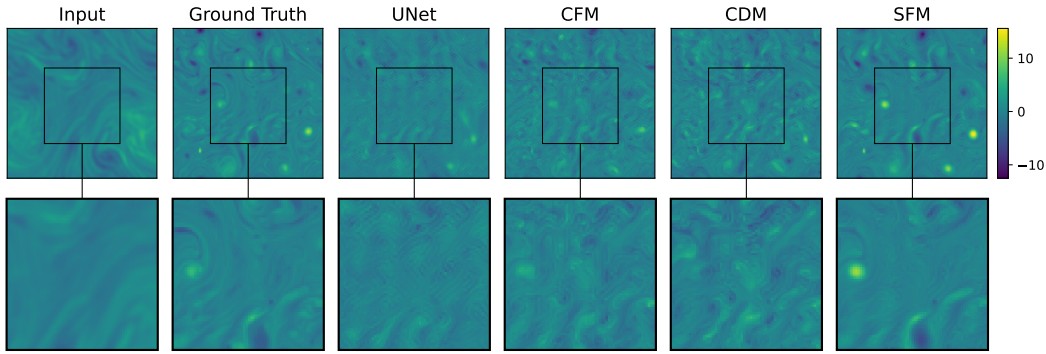

Figure 4: **AFM vs. Baselines for Kolmogorov Flow and** τ = 10: when the figures are zoomed in, it is apparent that AFM aligns closer to the ground truth, and the presence of high-frequency artifacts in the baseline models becomes more noticeable.

by steady-state forcing. The simulation uses a pseudo-spectral method on a $512 \times 512$ grid with a 3rd-order Adams-Bashforth time stepper. Different τ values (3, 5, 10) are used to generate training and test sets where higher values of τ looser coupling which translates to higher misalignment between low and high res simulations. Detailed descriptions of the equations and parameters are provided in Appendix A.4.2.

### 5.2.1 RESULTS

AFM consistently outperforms other methods across various skill metrics for different degrees of misalignment, denoted by τ (Table 3). Interestingly, while CDM appears to be the most calibrated, AFM demonstrates superior performance overall. In this case study, we use a $1 \times 1$ convolutional architecture for the AFM encoder. The ablations are deffered to Appendix B.5 due to limited space. This advantage is further supported by the spectral analysis in Fig. 12, where AFM's spectra most closely align with the ground truth, highlighting its robustness in preserving physical structures even under significant misalignment.

## 6 CONCLUSION

We introduced Adaptive Flow Matching (AFM) to address misaligned data in atmospheric downscaling tasks. AFM combines deterministic encoding of large-scale dynamics with Adaptive Flow Matching in latent space, effectively capturing both deterministic and stochastic components of the data. Experiments on synthetic and real-world datasets demonstrated that AFM outperforms existing methods, especially when input and target distributions are significantly misaligned.

A limitation of AFM is its reliance on paired datasets, which may not always be available. Future work includes extending AFM to handle unpaired or semi-supervised data and incorporating physical constraints to enhance the physical consistency of the outputs. Our work could also benefit from a theoretical analysis of its convergence properties, especially in relation to the adaptive noise scaling mechanism. Other meaningful approaches worth pursuing is the extension to other domains such as image-to-image translation, and alternative stochastic encoders like VAEs.

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

# A  APPENDIX

## A.1  PROOF OF PROPOSITION 1

**Proposition 1**. *For the perturbation model* $\mathbf{x}_t = \mathbf{x} + \sigma_t \boldsymbol{e} + \sigma_t \boldsymbol{\epsilon}$, *where the noise standard deviation is given by* $\sigma_t := (1-t)\sigma_z$, *the residual error by* $\boldsymbol{e} := (\mathcal{E}(\mathbf{y}) - \mathbf{x})/\sigma_z$, *and the noise* $\boldsymbol{\epsilon} \sim \mathcal{N}(0,1)$, *the flow matching for joint training of the encoder and flow reduces to the denoising objective in eq.* (10):

$$\min_{\mathcal{E},\boldsymbol{\theta}} \mathbb{E}_{\mathbf{x},\mathbf{y},\sigma_t \sim \mathcal{U}[0,\sigma_z]} \left[ (\sigma_z/\sigma_t)^2 \left\| \mathcal{D}_{\boldsymbol{\theta}}(\mathbf{x}_t, \sigma_t) - \mathbf{x} \right\|^2 \right]. \tag{17}$$

**Proof**. Consider the linear interpolant connecting $\mathbf{z} \sim \mathcal{N}(\mathcal{E}(\mathbf{y}), \sigma_z \mathbf{I})$ to the target distribution $p(\mathbf{x})$. This process can be expressed as:

$$\mathbf{x}_t = (1-t)(\mathcal{E}(\mathbf{y}) + \sigma_z \boldsymbol{\epsilon}) + t\mathbf{x} \tag{18}$$
$$= (1-t)\mathcal{E}(\mathbf{y}) + t\mathbf{x} + (1-t)\sigma_z \boldsymbol{\epsilon} \tag{19}$$
$$= (1-t)(\mathcal{E}(\mathbf{y}) - \mathbf{x}) + \mathbf{x} + (1-t)\sigma_z \boldsymbol{\epsilon} \tag{20}$$
$$= \mathbf{x} + (1-t)\sigma_z(\boldsymbol{e} + \boldsymbol{\epsilon}), \tag{21}$$

where $\boldsymbol{e} = (\mathcal{E}(\mathbf{y}) - \mathbf{x})/\sigma_z$.

Define $\sigma_t = (1-t)\sigma_z$ and reinterpret time $t$ in terms of the noise level $\sigma_t$, as is typical in continuous noise sampling (e.g., EDM). The perturbation kernel then becomes $\mathbf{x}_t = \mathbf{x} + \sigma_t(\boldsymbol{e} + \boldsymbol{\epsilon})$.

Now, consider the velocity training objective in (8):

$$\min_{\mathcal{E},\boldsymbol{\theta}} \mathbb{E}_{t,\mathbf{x},\mathbf{z} \sim \mathcal{N}(\mathcal{E}(\mathbf{y}),\sigma_z)} \left[ \left\| \boldsymbol{\nu}_\theta(\mathbf{x}_t, t) - (\mathbf{x} - \mathbf{z}) \right\|^2 \right].$$

where the velocity field expressed as a a denoiser (or equivalently score form Vincent (2011)):

$$\boldsymbol{\nu}_\theta(\mathbf{x}_t, t) = \frac{\mathcal{D}_{\boldsymbol{\theta}}(\mathbf{x}_t, \sigma_t) - \mathbf{x}_t}{1 - t}, \tag{22}$$

which is the reverse-time version of eq. (6), since in flow matching, time progresses from $t = 0$ to $t = 1$. Substituting $\boldsymbol{\nu}_\theta(\mathbf{x}_t, t)$ into the flow matching objective (8) and simplifying using $\mathbf{x}_t = (1-t)\mathbf{z} + t\mathbf{x}$ and $\sigma_t/\sigma_z = 1 - t$, we obtain the denoising objective in (17), completing the proof.

## A.2  NETWORK ARCHITECTURE AND TRAINING

For diffusion model training and sampling, we use EDM Karras et al. (2022), a continuous-time diffusion model available with a public codebase. EDM provides a physics-inspired design based on ODEs, auto-tuned for our scenario (see Table 1 in Karras et al. (2022)). We adopt most of the hyperparameters from EDM and make modifications as listed below.

**Architecture**: To cover the large field-of-view $448 \times 448$, we adapt the UNet from Song & Ermon (2019) by expanding it to 5 encoder and 5 decoder layers. The base channel size is 32, multiplied by [1, 2, 2, 4, 4] across layers. Attention resolution is set to 28. Time representation is handled via positional embedding, though this is disabled in the regression network, as no probability flow ODE is involved. No data augmentation is applied. The UNet has 12 million parameters, and we add 4 channels for sinusoidal positional embedding to improve spatial consistency, following practices in Dosovitskiy (2020); Carion et al. (2020). For the encoder $\mathcal{E}$, we use two architectures: (1) a simple $1 \times 1$ convolution layer, and (2) a UNet similar to the diffusion UNet but without time embedding. The same UNet is used for the regression network in CorrDiff.

**Optimizer**: We use the Adam optimizer with a learning rate of $10^{-4}$, $\beta_1 = 0.9$, $\beta_2 = 0.99$, and an exponential moving average (EMA) rate of $0.5$. Dropout is applied with a rate of $0.13$. Hyperparameters follow the guidelines in EDM Karras et al. (2022).

**Noise Schedule**: For AFM and CFM, we use a continuous noise schedule sampled uniformly $\sigma \in \mathcal{U}[0, \sigma_z]$. For CDM and CorrDiff, we use EDM's optimized log-normal noise schedule, $\sigma \sim \text{lognormal}(-1.2, 1.2)$.

**Training**: The regression network receives 12 input channels from the ERA5 data, while diffusion training concatenates these 12 input channels with 4 noise channels. EDM randomly selects noise

variance aiming to denoise samples per mini-batch. CFM, CDM, CFM and CorrDiff are trained for 50 million steps, whereas the regression UNet is trained for 20 million steps. Training is distributed across 8 DGX nodes, each with 8 A100 GPUs, using data parallelism and a total batch size of 512.

**Sampling**: Our sampling process employs Euler integration with 50 steps across all methods. We begin with a maximum noise variance $\sigma_{\text{max}}$ and decrease it to a minimum of $\sigma_{\text{min}} = 0.002$. The value of $\sigma_{\text{max}}$ varies depending on the method: for CDM and CorrDiff, we use $\sigma_{\text{max}} = 800$, as per the original implementation in Mardani et al. (2023); for CFM, we set $\sigma_{\text{max}} = 1$, as specified in Lipman et al. (2022); and for AFM, we use the $\sigma_z$ value learned during training.

## A.3 EVALUATION METRICS

### A.3.1 RMSE

The Root Mean Square Error (RMSE) is a standard evaluation metric used to measure the difference between the predicted values and the true values Chai & Draxler (2014). In the context of our problem, let $\mathbf{x}$ be the true target and $\hat{\mathbf{x}}$ be the predicted value. The RMSE is defined as:

$$\text{RMSE} = \sqrt{\mathbb{E}\left[\|\mathbf{x} - \hat{\mathbf{x}}\|^2\right]}. \tag{23}$$

This metric captures the average magnitude of the residuals, i.e., the difference between the predicted and true values. A lower RMSE indicates better model performance, as it suggests the predicted values are closer to the true values on average. RMSE is sensitive to large errors, making it an ideal choice for evaluating models where minimizing large deviations is critical.

### A.3.2 CRPS

The Continuous Ranked Probability Score (CRPS) is a measure used to evaluate probabilistic predictions Wilks (2011). It compares the entire predicted distribution $F(\hat{\mathbf{x}})$ with the observed data point $\mathbf{x}$. For a probabilistic forecast with cumulative distribution function (CDF) $F$, and the true value $\mathbf{x}$, the CRPS is given by:

$$\text{CRPS}(F, \mathbf{x}) = \int_{-\infty}^{\infty} \left(F(y) - \mathbb{I}(y \geq \mathbf{x})\right)^2 \, dy, \tag{24}$$

where $\mathbb{I}(\cdot)$ is the indicator function. Unlike RMSE, CRPS provides a more comprehensive evaluation of both the location and spread of the predicted distribution. A lower CRPS indicates a better match between the forecast distribution and the observed data. It is especially useful for probabilistic models that output a distribution rather than a single point prediction.

When applying CRPS to a finite ensemble of size $m$ approximating $F$ with the empirical CDF incurs an $O(1/m)$ bias favoring models with less spread. For small $m$ unbiased versions of the formulas should be used instead (Zamo & Naveau, 2018), but for the ensemble sizes here this is a small effect, so we used the more common biased formulas.

### A.3.3 SPREAD SKILL RATIO

The Spread-Skill Ratio (SSR) evaluates the reliability of the predicted uncertainty by comparing the spread (variance) of the predicted distribution with the accuracy of the predictions Gneiting & Raftery (2004). Let $\sigma_{\hat{\mathbf{x}}}$ be the standard deviation of the predicted distribution and RMSE as defined above. The SSR is defined as:

$$\text{SSR} = \frac{\sigma_{\hat{\mathbf{x}}}}{\text{RMSE}}. \tag{25}$$

An SSR value close to 1 indicates that the predicted uncertainty (spread) is well-calibrated with the model's predictive skill. If the SSR is less than 1, the model underestimates uncertainty, while an SSR greater than 1 indicates that the model overestimates uncertainty. This metric is particularly useful in evaluating the quality of probabilistic forecasts in terms of their sharpness (spread) and accuracy (skill).

| Description | Input | Output |
|---|---|---|
| Pixel Size | $36 \times 36$ | $448 \times 448$ |
| Single-Level Channels | Total Column Water Vapor
Temperature at 2 Meters
East Wind at 10 Meters
North Wind at 10 Meters | Maximum Radar Reflectivity
Temperature at 2 Meters
East Wind at 10 Meters
North Wind at 10 Meters |
| Pressure-Level Channels | Temperature
Geopotential
East Wind
North Wind | -
-
-
- |

Table 4: **ERA5-CWA Variables**: Input and target variables for the ERA5 to CWA downscaling task include both single-level and pressure-level variables, the latter at 850 and 500 hPa.

### A.4    FURTHER DETAILS OF THE DATASETS

Further details and visualizations of the ERA5-CWA and KF dataset, used throughout the paper, is presented here.

#### A.4.1    ERA5-CWA DATASET

Table table 4 summarizes the input-output channels and the corresponding resolutions. It is evident that the input and output channels generally differ, and even those that do overlap, such as (Temperature, East Wind, North Wind), are not perfectly aligned. For instance, comparing the Eastward Wind (10m) in the contour plots reveals the eye of the typhoon located northeast of the Taiwan region; see Fig. 5. Notably, the typhoon's eye shifts in the output due to the datasets originating from two different simulations, which solve distinct sets of partial differential equations (PDEs) at significantly different resolutions, resulting in divergent trajectories.

**Input Data (ERA5)**. This data for this study are derived from the ERA5 reanalysis, which provides a comprehensive set of atmospheric variables at various vertical levels Hersbach et al. (2020). For our analysis, we selected a subset of 12 variables. These include four variables (temperature, East and North components of the horizontal wind vector, and geopotential height) at two pressure levels (500 hPa and 850 hPa). Additionally, we incorporated single-level variables: 2-meter temperature, 10-meter wind vector components, and total column water vapor.

**Target Data (CWA)**. The horizontal range of these data encompasses a $900 \times 900$-km region containing Taiwan, with a horizontal resolution of approximately many output variables. We focus on four variables, three common to the input data – surface temperature and ruface horizontal wind components – and one of which is distinctly related to precipitating hydrometeors, the composite synthetic radar reflectivity at time of data assimilation. We represent the high-dimensional target data as $\mathbf{x} \in \mathbb{R}^{H \times W \times C}$, where $H = W = 448$ and $C = 4$. See table 4.

The dataset encompasses approximately four years (2018-2021) of observations, sampled at hourly intervals. For model development and validation, the data were partitioned chronologically. The training set comprises observations from 2018 to 2020, totaling 24,601 data points. The remaining data from 2021, consisting of 6,501 data points, were reserved for evaluation purposes. This temporal split allows for an assessment of the model's performance on future, unseen data, simulating real-world application scenarios.

Lastly, we upsample the input data to a $448 \times 448$ grid using bilinear interpolation to match the output resolution, a common practice with residual networks for consistency Hu et al. (2019); Zhang et al. (2018) (see Figure 5b for an example of input vs. target misalignment).

#### A.4.2    KOLMOGOROV-FLOW DATASET

A representative input-output sample of the KF dataset is shown in Fig. 6 for different misalignment degrees $\tau$.

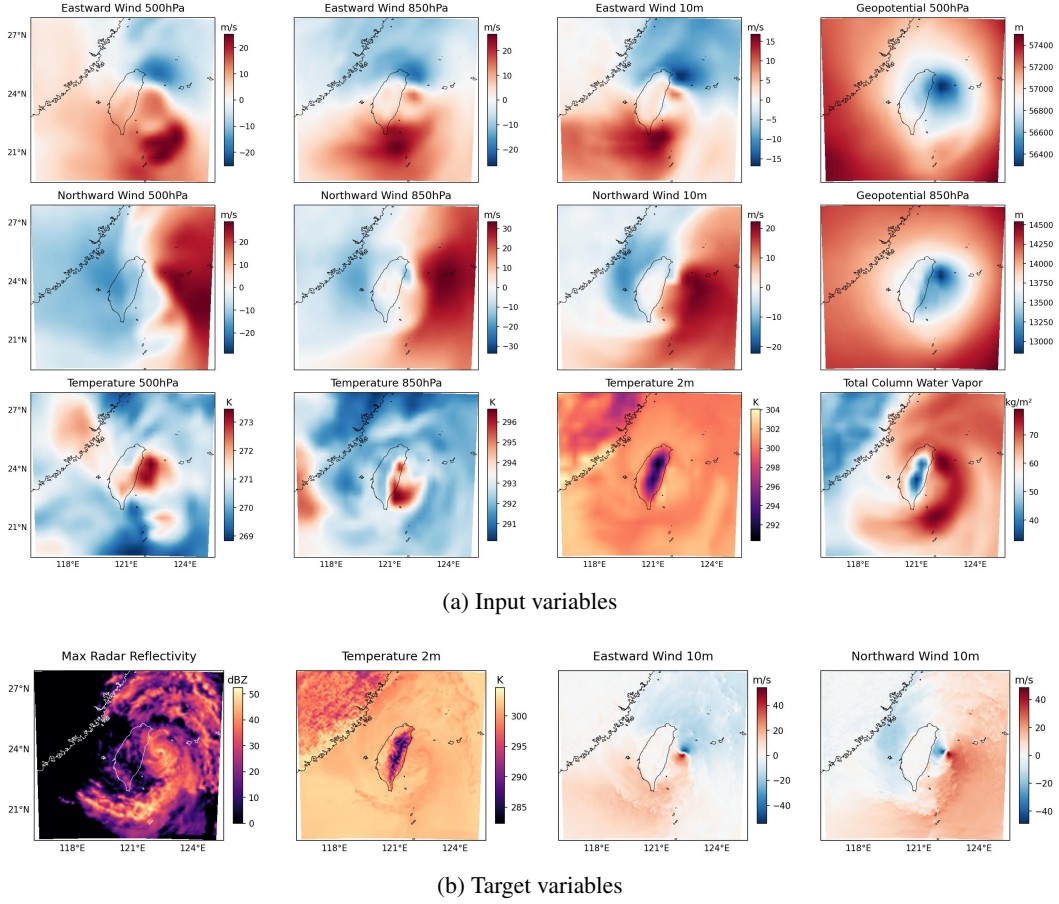

(a) Input variables

(b) Target variables

Figure 5: **Visualization of ERA5-CWA Dataset Variables.** The top row shows input variables such as temperature and wind at coarse resolution, while the bottom row presents the corresponding fine-resolution target variables. The maximum radar reflectivity is absent from the input variables and must be constructed by the model. This key misalignment between the low- and high-resolution data increases the complexity of the problem beyond standard super-resolution tasks.

**Dataset description**. We construct a toy dataset by simulating the dynamics given by:

$$
\begin{aligned}
\zeta_h + J(\psi_h, \zeta_h) &= F + \nu_h \nabla^7 \zeta_l - \zeta_l \tau_r^{-1} \\
\zeta_l + J(\psi_l, \zeta_l) &= -\tau^{-1}(\zeta_l - \zeta_h) + \nu_l \nabla^7 \zeta_l - \zeta_l \tau_r^{-1} \\
\nabla^2 \psi_l &= \zeta_l \\
\nabla^2 \psi_h &= \zeta_h.
\end{aligned}
\tag{26}
$$

Here, $J(f, g) = f_x g_y - f_y g_x$ is the Jacobian operator. The stream function is related to the velocity field by $\nabla \psi = (-u, v)$, implying that $\nabla \psi \cdot (u, v) = 0$, so that velocity points along contours of the stream-function. $\zeta_{l,h}$ represents the vorticity.

The $\zeta_l$ field represents a coarse-resolution simulation nudged towards a high resolution $\zeta_h$. The parameter $\tau$ controls the coupling strength between the $\zeta_l$ and $\zeta_h$ fields. A steady-state forcing $F = 10 \cos(10x)$ injects energy into the small-scale field $\zeta_h$ but not the low resolution $\zeta_l$, mimicking the injection of energy by sub-grid processes like convection or flow over topography. Stronger dissipation $\nu_l \gg \nu_h$ is used to limit the effective resolution of the large-scale field. A small amount of Rayleigh damping $\tau_r = 100$ is added to limit the pile-up of energy at large scales.

These equations are solved using a standard pseudo-spectral method on the GPU. The 3rd-order Adams-Bashforth time stepper is used for all but the hyper-viscosity terms; for these stiff terms, we use an backward Euler time stepper. The resolution is $512 \times 512$ and the timestep $dt = 0.001$. A

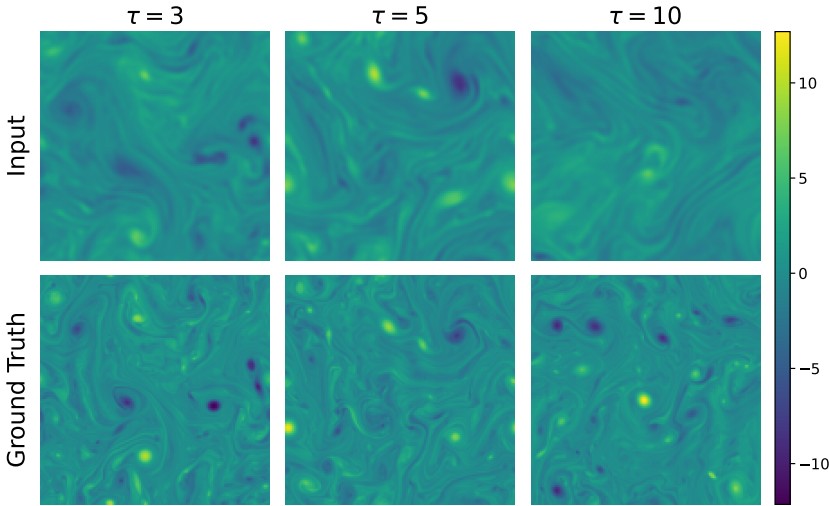

Figure 6: **Kolmogorov Flow Dataset.** Visualization of input and target Kolmogorov Flow dataset for varying levels of misalignment ($\tau = 3, 5, 10$). As $\tau$ increases, the discrepancy between the coarse and fine-resolution fields grows, offering a controlled environment to test downscaling performance.

$2/3$ de-aliasing filter is applied in spectral space every timestep (Orszag, 1971). Outputs are saved every $\delta = 0.2$ time units.

We create datasets for different $\tau$ values: $3, 5, 10, 20$. Higher $\tau$ corresponds to greater misalignment between the coarse and high-resolution simulations. This variation allows us to assess the robustness of our method across different levels of coupling and identify potential thresholds in $\tau$ beyond which certain downscaling approaches may become unreliable. For each $\tau$ value, we generate a dataset comprising $100,000$ training points and $500$ test points.

## B    ADDITIONAL EXPERIMENTS

We report additional experiments for both Taiwan CWA downscaling and KF downscaling, with a particular focus on the two-stage overfitting issues, ablation studies, analysis of the adaptive stochasticity $\sigma_z$ and ensemble analysis.

### B.1    ANALYSIS OF CORRDIFF OVERFITTING ISSUES

Here we investigate the overfitting challenges inherent to the two-stage approach utilized by CorrDiff, focusing on the training dynamics of the UNet regression model.

As shown in Figure 7, the UNet regressor exhibits signs of overfitting after approximately 500k training steps. This is evident from the divergence between the training and validation MSE losses across all channels. We also note that the radar and temperature channels exhibit stronger overfitting.

Table 5 further elucidates the impact of the number of training steps on CorrDiff's performance. Models trained for a limited number of steps (e.g., 0.5M) demonstrate better calibration and higher stochasticity, as reflected by higher SSR values. Conversely, models subjected to extensive training (e.g., 50M steps) show diminished diversity in their predictions and lower SSR values. This reduction in SSR suggests that the regression component becomes overly confident, producing residuals that are narrowly concentrated around zero. Subsequently the diffusion model has limited variability to model, which diminishes its ability to generate meaningful corrections at test-time.

These observations highlight a critical limitation of CorrDiff's two-stage methodology: the initial regression stage is prone to overfitting, which in turn constrains the diffusion model's capacity to generalize effectively. To address this issue, the authors of CorrDiff use early-stopping to chose a less overfit UNet.

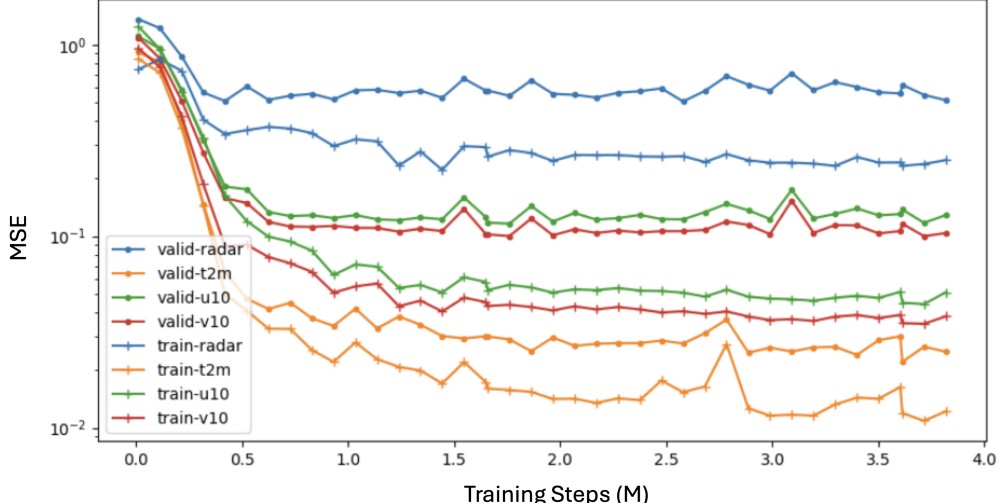

Figure 7: **UNet train and validation loss during training on CWA** $448 \times 448$ **data.** Evolution of training and validation MSE for the UNet regressor across training steps. We observe that the UNet starts overfitting as early as 500k steps. Furthermore, certain variables like radar reflectivity(blue) and temperature(yellow) show higher amounts of overfitting. This behavior is the reason that two-stage approaches like CorrDiff utilize early stopping to avoid overfitting the training data, because this is very difficult to correct on the second stage. AFM resolves this issue by leveraging end-to-end training determinsticn loss weghting as well as adaptive noise scaling.

| Variable | Metric | UNet training steps | | |
| | | 0.5M | 2M | 50M |
|---|---|---|---|---|
| **Radar** | **RMSE** ↓ | 5.08 | 5.28 | 5.13 |
| | **CRPS** ↓ | 1.81 | 1.89 | 2.10 |
| | **SSR** $\rightarrow$ 1 | 0.52 | 0.35 | 0.14 |
| **Eastward Wind** | **RMSE** ↓ | 1.51 | 1.53 | 1.48 |
| | **CRPS** ↓ | 0.83 | 0.89 | 0.87 |
| | **SSR** $\rightarrow$ 1 | 0.60 | 0.42 | 0.39 |
| **Northward Wind** | **RMSE** ↓ | 1.72 | 1.70 | 1.65 |
| | **CRPS** ↓ | 0.97 | 1.01 | 0.99 |
| | **SSR** $\rightarrow$ 1 | 0.56 | 0.39 | 0.37 |
| **Temperature** | **RMSE** ↓ | 0.96 | 0.93 | 0.91 |
| | **CRPS** ↓ | 0.57 | 0.58 | 0.53 |
| | **SSR** $\rightarrow$ 1 | 0.41 | 0.28 | 0.31 |

Table 5: **Impact of disjoint regression training on CorrDiff.** This table demonstrates how varying the number of training steps for the UNet regression in the first stage affects the final skills of CorrDiff. UNet checkpoints trained for 0.5M, 2M, and 50M steps were utilized to train different diffusion models in the second stage. The results show that less-trained UNet models are better calibrated and exhibit superior stochasticity in their outcomes (SSR $\rightarrow$ 1). In contrast, increased training leads to less diverse ensembles, suggesting that the model struggles to correct a biased UNet while maintaining variability in its results.

In contrast, our proposed method, AFM, addresses these overfitting issues by adopting an end-to-end training paradigm. By balancing the deterministic and stochastic losses and taking into account the different stochasticity levels between the variables with the adaptive noise scaling, it treads the overfitting issue in a more principled and effective way.

| Encoder | Channel Multipliers | Number of Parameters |
|---------|---------------------|----------------------|
| L | [1, 2, 2, 4, 4] | 12M |
| M | [1, 2, 2, 2, 2] | 5M |
| S | [1, 1, 2, 2, 2] | 1M |
| XS | [1, 1, 1, 2, 2] | 0.2M |
| $1 \times 1$ Conv. | - | 60 |

Table 6: Details of different encoder sizes used in the experiments. For the UNet the channel multipliers are applied to the base channel size of 32 at each layer.

### B.2 ANALYSIS OF ADAPTIVE $\sigma_{\mathbf{z}}$ DURING TRAINING

To evaluate the behavior of the adaptive noise scaling mechanism in our Adaptive Flow Matching (AFM) model, we monitored the sigma values across different channels during the training process for the model with $\lambda = 0.25$ as depicted in Figure 8. The sigma values are initially set to 1 for all channels. During the early stages of training, sigma increases across the channels, due to the high encoder error. As training progresses and the encoder's performance improves, the sigma values begin to stabilize and converge towards their final values.

Notably, the radar reflectivity channel exhibits the highest sigma values throughout the training process, reflecting its inherently stochastic nature. This is consistent with our understanding that radar data contains significant variability and uncertainty. In contrast, the temperature channel consistently shows the lowest sigma values, aligning with its more deterministic characteristics. These variations in sigma across channels underscore the effectiveness of our adaptive noise scaling approach, as it allows the model to appropriately adjust noise levels based on the inherent uncertainty of each channel. This adaptability is crucial for managing misaligned data with differing degrees of stochasticity, thereby enhancing the overall performance and reliability of the AFM model in multiscale physics applications.

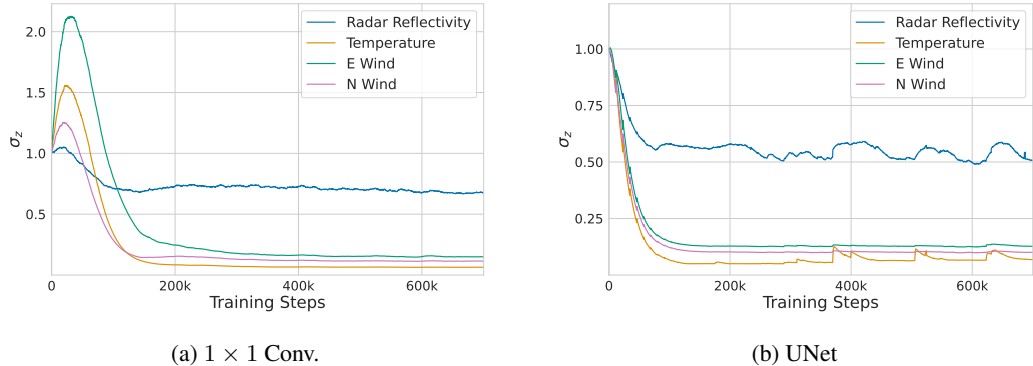

(a) $1 \times 1$ Conv.

(b) UNet

Figure 8: **Adaptive $\sigma_{\mathbf{z}}$ values over training steps for different channels.** The plot corresponds to the AFM model using $1 \times 1$ Conv. (a) and UNet (b) encoders with $\lambda = 0.25$. With $1 \times 1$ convolution, $\sigma_z$ increases during early training due to high encoder error and subsequently converges as the encoder improves. With UNet, $\sigma_z$ starts decreasing with training. In both cases, radar reflectivity exhibits the highest $\sigma$, indicating its stochastic nature, while temperature shows the lowest, reflecting its deterministic characteristics. The varying sigma values across channels demonstrate why adaptive noise scaling is effective in managing misaligned data with differing levels of stochasticity.

### B.3 CWA ABLATION STUDIES

To evaluate the effectiveness of the AFM model and understand the impact of its components, we conducted ablation studies on the CWA weather downscaling task at $112 \times 112$ resolution. We focused on varying the $\lambda$ parameter, different encoder types, the use of adaptive $\sigma_{\mathbf{z}}$, and $\mathbf{y}$ conditioning. The results are summarized in Tables 7, 8, and 9.

### B.3.1 EFFECT OF $\lambda$ PARAMETER AND ENCODER TYPE

Table 7 presents the performance of the AFM model with different $\lambda$ values and encoder types. The $\lambda$ parameter controls the trade-off between data fitting and uncertainty regularization in the AFM model.

$1 \times 1$ **Conv. Encoder:** For the $1 \times 1$ Conv. encoder, setting $\lambda = 0$ achieves the best performance across most variables, particularly for radar reflectivity, eastward wind, and northward wind. This indicates that the model benefits from minimal regularization when using a simpler encoder, allowing it to focus on fitting the data closely. The low parameter count (only 60 parameters) of the $1 \times 1$ Conv. encoder might limit its ability to capture all the information in the low-resolution input, making less regularization advantageous.

**UNet Encoder:** In contrast, the UNet encoder shows improved performance with a small regularization parameter of $\lambda = 0.25$. This suggests that the more complex architecture of the UNet benefits from some regularization to prevent overfitting and to enhance its generalization capabilities. The regularization helps balance the model's capacity, leading to better accuracy and uncertainty estimates across multiple variables.

### B.3.2 IMPACT OF ADAPTIVE $\sigma_{\mathbf{z}}$

Table 8 examines the effect of enabling or disabling adaptive $\sigma_{\mathbf{z}}$ for both encoder types.

**Findings:** Enabling adaptive $\sigma_{\mathbf{z}}$ consistently enhances the model's performance across all variables and both encoder types. The improvements in RMSE, CRPS, and SSR metrics suggest that adaptive $\sigma_{\mathbf{z}}$ allows the model to better capture the underlying uncertainty in the data. This adaptive approach provides the flexibility to model variable levels of uncertainty across different regions and variables, leading to more accurate and reliable predictions.

### B.3.3 EFFECT OF $\mathbf{y}$ CONDITIONING

Table 9 assesses the impact of enabling or disabling $\mathbf{y}$ conditioning in the AFM model for both encoders.

$1 \times 1$ **Conv. Encoder:** Disabling $\mathbf{y}$ conditioning provides better results across most metrics for this encoder. Given the simplicity of the $1 \times 1$ Conv. encoder and its limited parameter count, it may not effectively utilize the additional information provided by $\mathbf{y}$ conditioning. The model performs better when focusing on directly mapping the input to the output without the added complexity.

**UNet Encoder:** Enabling $\mathbf{y}$ conditioning yields the best results for the UNet encoder. The more complex architecture of the UNet can leverage the additional context from $\mathbf{y}$ conditioning to improve its predictions. This demonstrates the capacity of the UNet encoder to capture and utilize supplementary information, enhancing both accuracy and uncertainty estimation.

### B.3.4 SUMMARY

These ablation studies highlight the strengths of the AFM model and its components in weather downscaling tasks:

- **Effectiveness of Adaptive $\sigma_{\mathbf{z}}$:** Enabling adaptive $\sigma_{\mathbf{z}}$ consistently improves model performance across both encoder types and all variables. This underscores the importance of modeling spatially varying uncertainty in complex weather data.
- **Encoder Choice and Regularization:** The $1 \times 1$ Conv. encoder performs best without regularization ($\lambda = 0$), indicating that minimal regularization benefits simpler models. For the UNet encoder, a small regularization parameter ($\lambda = 0.25$) yields better results, suggesting that regularization helps prevent overfitting in more complex models.
- **Impact of $\mathbf{y}$ Conditioning:** $\mathbf{y}$ conditioning enhances performance for the UNet encoder but not for the $1 \times 1$ Conv. encoder. This suggests that the effectiveness of incorporating additional context depends on the model's capacity to utilize that information.

Overall, the AFM model demonstrates strong performance and flexibility in weather downscaling tasks. By carefully selecting model components such as encoder type, adaptive $\sigma_{\mathbf{z}}$, and regulariza-

tion parameter $\lambda$, the AFM can be tailored to balance predictive accuracy and uncertainty estimation effectively. These findings highlight the potential of the AFM model as a powerful tool for probabilistic weather downscaling.

| Encoder | $\lambda$ | Radar | | | Temperature | | | Eastward Wind | | | Northward Wind | | |
|---|---|---|---|---|---|---|---|---|---|---|---|---|---|
| | | RMSE ↓ | CRPS ↓ | SSR → 1 | RMSE ↓ | CRPS ↓ | SSR → 1 | RMSE ↓ | CRPS ↓ | SSR → 1 | RMSE ↓ | CRPS ↓ | SSR → 1 |
| $1 \times 1$ Conv. | 0.00 | **4.82** | **1.66** | **0.72** | 0.99 | 0.59 | **0.47** | **1.42** | **0.78** | **0.68** | **1.58** | **0.89** | **0.66** |
| | 0.25 | 5.10 | 1.83 | 0.42 | **0.85** | **0.50** | 0.43 | 1.46 | 0.84 | 0.46 | 1.63 | 0.92 | 0.46 |
| | 1.00 | 5.04 | 1.83 | 0.40 | 1.05 | 0.60 | 0.34 | 1.50 | 0.88 | 0.44 | 1.67 | 0.95 | 0.44 |
| UNet | 0.00 | 5.06 | 1.83 | 0.38 | 1.01 | 0.57 | 0.36 | **1.48** | **0.85** | 0.46 | 1.64 | 0.93 | 0.46 |
| | 0.25 | **4.95** | **1.78** | **0.44** | **0.94** | **0.54** | **0.40** | 1.49 | 0.85 | **0.53** | **1.58** | **0.95** | **0.50** |
| | 1.00 | 5.11 | 1.89 | 0.41 | 1.07 | 0.62 | 0.36 | 1.53 | 0.91 | 0.41 | 1.74 | 1.03 | 0.40 |

Table 7: **Encoder and $\lambda$ Ablations on CWA $112 \times 112$ Dataset.** For this table we keep the best configurations for each $\lambda$ value based on radar reflectivity. We separate per $1 \times 1$ Conv. and UNet encoder to elucidate differences. Best results for each encoder are highlighted in bold. No regularization ($\lambda = 0$) works best for the $1 \times 1$ Conv. encoder, while for UNet, a small $\lambda = 0.25$ value yields the best results across multiple variables. Overall the $1 \times 1$ Conv. without regularization is the best configuration.

| Encoder | Adapt $\sigma_{\mathbf{z}}$ | Radar | | | Temperature | | | Eastward Wind | | | Northward Wind | | |
|---|---|---|---|---|---|---|---|---|---|---|---|---|---|
| | | RMSE ↓ | CRPS ↓ | SSR → 1 | RMSE ↓ | CRPS ↓ | SSR → 1 | RMSE ↓ | CRPS ↓ | SSR → 1 | RMSE ↓ | CRPS ↓ | SSR → 1 |
| $1 \times 1$ Conv. | ✗ | 5.01 | 1.88 | 0.45 | 1.13 | 0.66 | 0.39 | 1.50 | 0.86 | 0.48 | 1.64 | 0.94 | 0.47 |
| | ✓ | **4.82** | **1.66** | **0.72** | **0.99** | **0.59** | **0.47** | **1.42** | **0.78** | **0.68** | **1.58** | **0.89** | **0.66** |
| UNet | ✗ | 5.06 | 1.83 | 0.38 | 1.01 | 0.57 | 0.36 | 1.48 | 0.85 | 0.46 | 1.64 | 0.93 | 0.46 |
| | ✓ | **4.95** | **1.78** | **0.44** | **0.94** | **0.54** | **0.40** | **1.49** | **0.85** | **0.53** | **1.58** | **0.95** | **0.50** |

Table 8: **Adaptive $\sigma_{\mathbf{z}}$ Ablation on CWA $112 \times 112$ Dataset for $1 \times 1$ Conv. and UNet Encoders.** This table examines the effect of enabling (✓) or disabling (✗) Adaptive $\sigma_{\mathbf{z}}$ across both $1 \times 1$ Conv. and UNet encoders. Best results for each encoder and metric are highlighted in bold. Results indicate that the proposed adaptive $\sigma_z$ consistently improve results.

| Encoder | y cond. | Radar | | | Temperature | | | Eastward Wind | | | Northward Wind | | |
|---|---|---|---|---|---|---|---|---|---|---|---|---|---|
| | | RMSE ↓ | CRPS ↓ | SSR → 1 | RMSE ↓ | CRPS ↓ | SSR → 1 | RMSE ↓ | CRPS ↓ | SSR → 1 | RMSE ↓ | CRPS ↓ | SSR → 1 |
| $1 \times 1$ Conv. | ✗ | **4.82** | **1.66** | **0.72** | 0.99 | 0.59 | **0.47** | **1.42** | **0.78** | **0.68** | 1.63 | **0.89** | **0.66** |
| | ✓ | 5.06 | 1.82 | 0.34 | **0.89** | **0.54** | 0.40 | 1.46 | 0.82 | 0.45 | **1.59** | 0.92 | 0.46 |
| UNet | ✗ | 5.06 | 1.83 | 0.38 | 1.01 | 0.57 | 0.36 | 1.48 | **0.85** | 0.46 | **1.64** | 0.93 | 0.46 |
| | ✓ | **4.95** | **1.78** | **0.44** | **0.94** | **0.54** | **0.40** | 1.49 | 0.85 | **0.53** | 1.67 | **0.95** | **0.50** |

Table 9: **y conditioning ablation on CWA $112 \times 112$ Dataset for $1 \times 1$ Conv. and UNet Encoders.** This table assesses the impact of enabling (✓) or disabling (✗) **y** conditioning across both $1 \times 1$ Conv. and UNet encoders. Best results for each encoder and metric are highlighted in bold. Results indicate that for the $1 \times 1$ encoder the conditioning is beneficial while for UNet the opposite stands. This makes sense since the $1 \times 1$ Conv. encoder has only 60 parameters and might not capture all the information in the low-resolution input.

| Model | $\lambda$ | Adapt. $\sigma_{max}$ | Use $\mathbf{x}_{low}$ | Radar RMSE ↓ | Radar CRPS ↓ | Radar SSR → 1 | Temp. RMSE | Temp. CRPS | Temp. SSR | East. RMSE | East. CRPS | East. SSR | North. RMSE | North. CRPS | North. SSR |
|---|---|---|---|---|---|---|---|---|---|---|---|---|---|---|---|
| AFM 1×1 Conv. | 0.00 | ✗ | ✗ | 5.16 | 1.88 | 0.44 | 1.12 | 0.65 | 0.36 | 1.53 | 0.88 | 0.47 | 1.72 | 0.98 | 0.46 |
| | | ✓ | ✗ | **4.82** | **1.66** | **0.72** | 0.99 | 0.59 | **0.47** | **1.42** | **0.78** | **0.68** | 1.63 | **0.89** | **0.66** |
| | | | ✓ | 5.06 | 1.82 | 0.34 | 0.87 | 0.52 | **0.44** | 1.44 | 0.81 | 0.49 | 1.60 | **0.89** | 0.49 |
| | 0.25 | ✗ | ✗ | 5.10 | 1.83 | 0.42 | **0.85** | **0.50** | 0.43 | 1.46 | 0.84 | 0.46 | 1.63 | 0.92 | 0.46 |
| | | ✓ | ✗ | 5.11 | 1.85 | 0.37 | 0.89 | 0.56 | 0.28 | 1.53 | 0.92 | 0.36 | 1.71 | 1.02 | 0.34 |
| | | | ✓ | 5.12 | 1.87 | 0.29 | 0.89 | 0.54 | 0.40 | 1.46 | 0.84 | 0.45 | 1.59 | 0.90 | 0.46 |
| | 1.00 | ✗ | ✗ | 5.04 | 1.83 | 0.40 | 1.05 | 0.60 | 0.34 | 1.50 | 0.88 | 0.44 | 1.67 | 0.95 | 0.44 |
| | | ✓ | ✗ | 5.11 | 1.85 | 0.36 | 0.92 | 0.58 | 0.30 | 1.52 | 0.92 | 0.35 | 1.71 | 1.01 | 0.34 |
| | | | ✓ | 5.12 | 1.88 | 0.30 | 0.86 | **0.50** | 0.41 | **1.43** | 0.82 | 0.46 | **1.58** | 0.90 | 0.45 |
| AFM UNet | 0.00 | ✗ | ✗ | 5.06 | 1.83 | 0.38 | 1.01 | 0.57 | 0.36 | 1.48 | 0.85 | 0.46 | 1.64 | 0.93 | 0.46 |
| | | ✓ | ✗ | 5.13 | 1.84 | 0.43 | **0.85** | 0.52 | 0.37 | **1.43** | **0.81** | 0.50 | 1.60 | 0.89 | 0.48 |
| | | | ✓ | 5.07 | 1.84 | 0.36 | **0.85** | 0.52 | 0.34 | 1.44 | 0.83 | 0.43 | 1.60 | 0.91 | 0.43 |
| | 0.25 | ✗ | ✗ | 5.01 | 1.88 | 0.45 | 1.13 | 0.66 | 0.39 | 1.50 | 0.86 | 0.48 | 1.64 | 0.94 | 0.47 |
| | | ✓ | ✗ | 5.01 | 1.82 | 0.32 | **0.85** | 0.52 | 0.31 | **1.43** | 0.85 | 0.40 | **1.58** | 0.93 | 0.39 |
| | | | ✓ | **4.95** | **1.78** | **0.44** | 0.94 | 0.54 | 0.40 | 1.49 | 0.85 | **0.53** | 1.67 | 0.95 | **0.50** |
| | 1.00 | ✗ | ✗ | 5.11 | 1.89 | 0.41 | 1.07 | 0.62 | 0.36 | 1.53 | 0.91 | 0.41 | 1.74 | 1.03 | 0.40 |
| | | ✓ | ✗ | 5.04 | 1.85 | 0.29 | **0.84** | 0.53 | 0.26 | 1.45 | 0.87 | 0.35 | 1.62 | 0.97 | 0.34 |
| | | | ✓ | 5.07 | 1.85 | 0.31 | 0.88 | 0.54 | 0.29 | 1.46 | 0.88 | 0.38 | 1.65 | 0.98 | 0.37 |

Table 10: **Complete ablation results for the CWA** $112 \times 112$ **dataset.** Best two models in each variable/metric in bold.

| | Model | CFM | CDM | UNet | AFM |
|---|---|---|---|---|---|
| **Radar** | **RMSE**↓ | 5.06 | 4.95 | 4.94 | **4.82** |
| | **CRPS**↓ | 1.84 | 1.74 | - | **1.66** |
| | **MAE**↓ | **2.41** | 2.49 | 2.45 | 2.63 |
| | **SSR** → 1 | 0.36 | 0.52 | - | **0.72** |
| **Temperature** | **RMSE** | 0.86 | 0.87 | 0.87 | 0.99 |
| | **CRPS** | **0.50** | 0.52 | - | 0.59 |
| | **MAE** | 0.64 | 0.64 | 0.64 | 0.74 |
| | **SSR** | 0.45 | 0.38 | - | **0.47** |
| **East. Wind** | **RMSE** | **1.42** | 1.44 | **1.42** | **1.42** |
| | **CRPS** | 0.81 | 0.81 | - | **0.78** |
| | **MAE** | **1.04** | 1.05 | 1.05 | 1.05 |
| | **SSR** | 0.48 | 0.49 | - | **0.68** |
| **North. Wind** | **RMSE** | **1.59** | **1.59** | 1.60 | 1.63 |
| | **CRPS** | **0.89** | **0.89** | - | **0.89** |
| | **MAE** | **1.14** | **1.14** | 1.16 | 1.19 |
| | **SSR** | 0.47 | 0.48 | - | **0.66** |

Table 11: **Performance Comparison of Models on CWA** $112 \times 112$ **Dataset.** The AFM model has a $1 \times 1$ Conv. encoder, $\lambda = 0$, adaptive $\sigma_z$ and no $y$ conditioning. Overall, the AFM model exhibits strong performance across different metrics and variables, particularly excelling in its calibration (variability). Best results for each metric are highlighted in bold. Note that for deterministic models, CRPS equals MAE.

| Variable | Metric | $\lambda$ | | | | | | |
|---|---|---|---|---|---|---|---|---|
| | | **0.0** | **0.01** | **0.1** | **0.25** | **0.5** | **1.0** | **2.5** |
| Radar | RMSE ↓ | **4.90** | 5.11 | 5.03 | 4.97 | 5.00 | 5.25 | 5.27 |
| | CRPS ↓ | **1.78** | 1.92 | 1.88 | 1.85 | 1.87 | 1.97 | 1.96 |
| | MAE ↓ | **2.42** | 2.66 | 2.47 | 2.47 | 2.47 | 2.59 | 2.75 |
| | SSR→ 1 | 0.44 | 0.49 | 0.39 | 0.42 | 0.41 | 0.39 | **0.52** |
| Temperature | RMSE ↓ | 1.00 | 1.00 | 0.87 | 0.86 | **0.82** | 1.00 | 0.85 |
| | CRPS ↓ | 0.52 | 0.62 | 0.54 | 0.52 | **0.48** | 0.56 | **0.48** |
| | MAE ↓ | 0.67 | 0.78 | 0.66 | 0.64 | **0.60** | 0.68 | 0.62 |
| | SSR → 1 | 0.47 | **0.48** | 0.38 | 0.40 | 0.41 | 0.32 | 0.45 |
| East. Wind | RMSE ↓ | **1.44** | 1.52 | 1.49 | 1.48 | 1.49 | 1.55 | 1.48 |
| | CRPS ↓ | **0.80** | 0.86 | 0.87 | 0.87 | 0.88 | 0.92 | 0.86 |
| | MAE ↓ | **1.07** | 1.13 | 1.09 | 1.09 | 1.09 | 1.14 | 1.08 |
| | SSR → 1 | **0.61** | 0.55 | 0.42 | 0.40 | 0.41 | 0.38 | 0.43 |
| North. Wind | RMSE ↓ | **1.61** | **1.61** | 1.66 | 1.67 | 1.67 | 1.72 | 1.65 |
| | CRPS ↓ | **0.88** | **0.88** | 0.95 | 0.96 | 0.96 | 1.00 | 0.95 |
| | MAE ↓ | **1.17** | **1.17** | 1.19 | 1.20 | 1.19 | 1.24 | 1.18 |
| | SSR → 1 | 0.58 | **0.61** | 0.41 | 0.41 | 0.40 | 0.38 | 0.43 |

Table 12: **AFM ablations for $\lambda$ on the CWA weather downscaling task at full resolution** $448 \times 448$. For this ablation, a $1 \times 1$ Conv. encoder was used, $\sigma_z$ was set to 1, and no $\mathbf{y}$ conditioning was employed. Overall, $\lambda = 0$ seems to produce better estimates except for temperature, whose deterministic nature benefits from the added regularization.

| Variable | Metric | Encoder | | | |
|---|---|---|---|---|---|
| | | **L** | **M** | **S** | **XS** |
| Radar | RMSE ↓ | **4.93** | 4.96 | 4.98 | 4.98 |
| | CRPS ↓ | **1.82** | 1.84 | 1.85 | 1.85 |
| | MAE ↓ | **2.44** | 2.52 | 2.52 | 2.46 |
| | SSR→ 1 | 0.40 | **0.43** | 0.42 | 0.39 |
| Temperature | RMSE ↓ | 1.01 | 1.00 | **0.99** | 1.02 |
| | CRPS ↓ | 0.55 | **0.54** | **0.54** | 0.56 |
| | MAE ↓ | **0.68** | **0.68** | **0.68** | 0.70 |
| | SSR → 1 | 0.34 | 0.38 | **0.40** | 0.38 |
| East. Wind | RMSE ↓ | 1.48 | 1.48 | **1.47** | 1.50 |
| | CRPS ↓ | 0.85 | **0.83** | **0.83** | 0.86 |
| | MAE ↓ | 1.10 | **1.08** | **1.08** | 1.11 |
| | SSR → 1 | 0.49 | **0.53** | **0.53** | 0.51 |
| North. Wind | RMSE ↓ | 1.64 | 1.64 | **1.63** | 1.66 |
| | CRPS ↓ | 0.92 | 0.92 | **0.91** | 0.93 |
| | MAE ↓ | 1.19 | 1.20 | **1.19** | 1.21 |
| | SSR → 1 | 0.48 | **0.51** | **0.52** | 0.50 |

Table 13: **AFM ablations for encoder size on the CWA weather downscaling task at full resolution** $448 \times 448$. For this ablation, we use $\lambda = 0$, $\sigma_z = 1$, and no $\mathbf{y}$ conditioning. Larger encoders improve performance for complex spatial data like radar, while for temperature and wind data, smaller encoders are adequate and sometimes even slightly better. The optimal encoder size depends on the specific variable being predicted but overall the difference are not significant.

| Variable | Metric | y conditioning | |
| --- | --- | --- | --- |
| | | ✗ | ✓ |
| Radar | **RMSE** ↓ | 5.09 | **4.90** |
| | **CRPS** ↓ | 1.88 | **1.80** |
| | **MAE** ↓ | 2.33 | **2.24** |
| | **SSR**→ 1 | 0.24 | **0.25** |
| Temperature | **RMSE** ↓ | 0.92 | **0.89** |
| | **CRPS** ↓ | 0.55 | **0.50** |
| | **MAE** ↓ | 0.67 | **0.64** |
| | **SSR** → 1 | 0.33 | **0.43** |
| East. Wind | **RMSE** ↓ | 1.49 | **1.45** |
| | **CRPS** ↓ | 0.91 | **0.86** |
| | **MAE** ↓ | 1.10 | **1.07** |
| | **SSR** → 1 | 0.34 | **0.41** |
| North. Wind | **RMSE** ↓ | 1.66 | **1.61** |
| | **CRPS** ↓ | 1.00 | **0.94** |
| | **MAE** ↓ | 1.20 | **1.18** |
| | **SSR** → 1 | 0.33 | **0.41** |

Table 14: **AFM ablations for $y$ conditioning for the CWA weather downscaling task at full resolution** $448 \times 448$. For this ablation, we use adaptive $\sigma_z$, $\lambda = 0.25$ and a UNet encoder guided by the ablations in the $112 \times 112$ resolution. Including **y** conditioning consistently improves performance across all metrics. RMSE, CRPS, and MAE are lower when **y** conditioning is used, and SSR values are closer to 1.

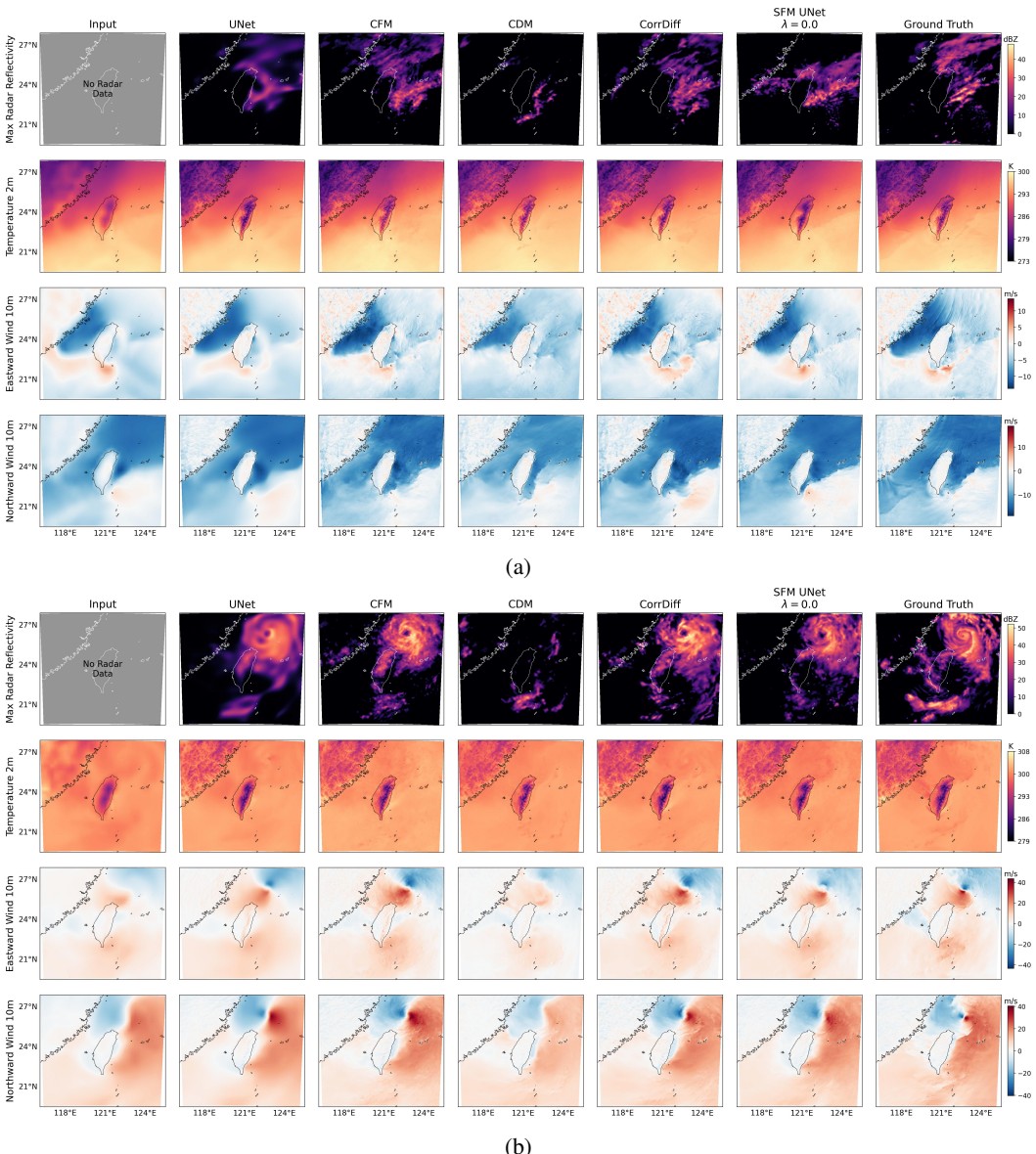

Figure 9: (Cont.)

## B.4 ENSEMBLE ANALYSIS

For a few representative samples, several ensemble members and the ensemble mean are shown in Figures 10a to 10e. The generated samples using different random seeds exhibit notable diversity, particularly for channels like Radar Reflectivity. This diversity confirms the model's ability to produce a well-dispersed ensemble, which is crucial for achieving a calibrated and reliable probabilistic forecast. Additionally, the ensemble mean closely aligns with the true target, indicating that the model successfully captures the underlying physical processes while preserving uncertainty across channels.

Animated PNGs of ensemble members for different models and $\tau$ are provided at `https://t.ly/ZCq9Z`.

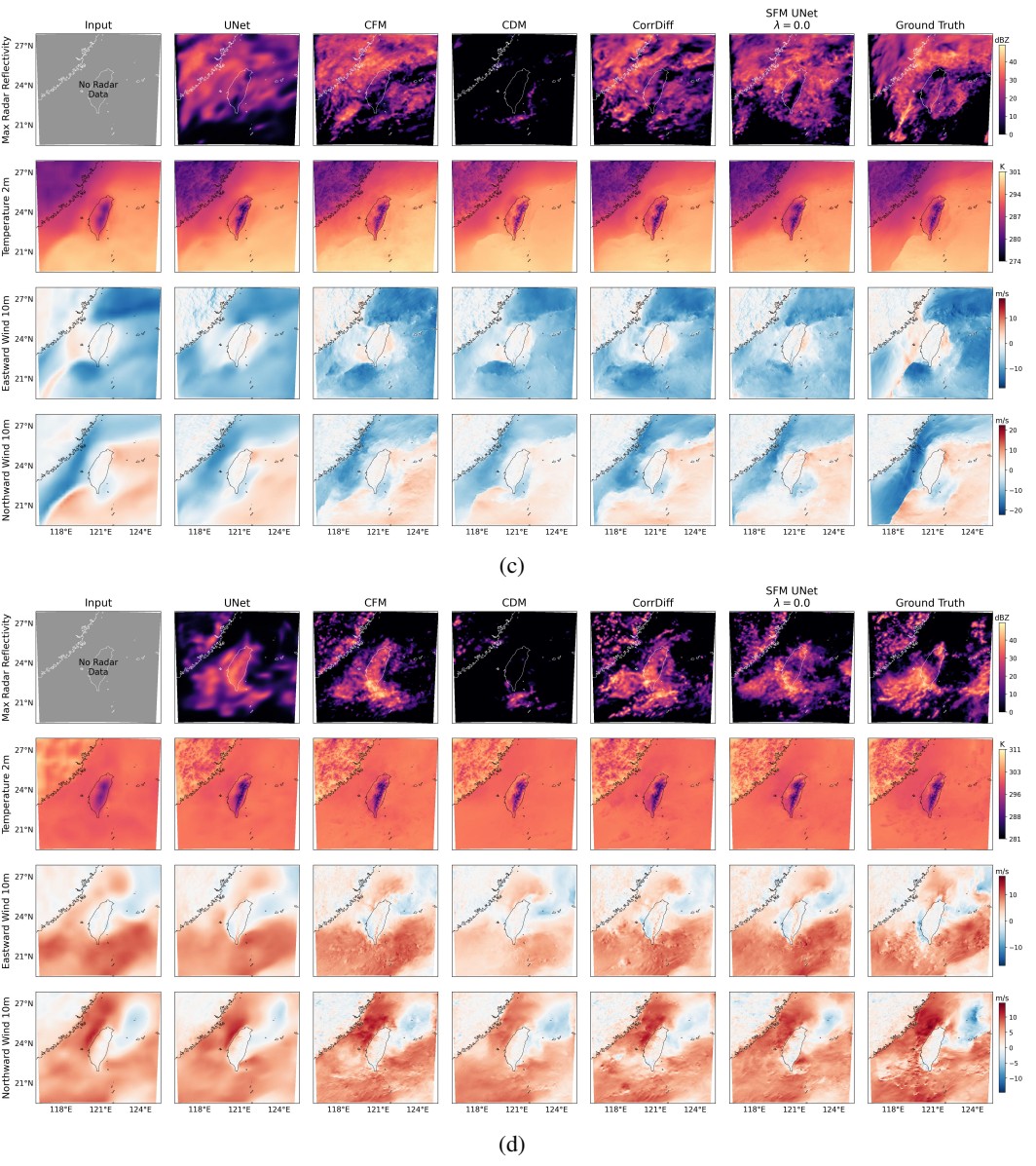

Figure 9: (Cont.)

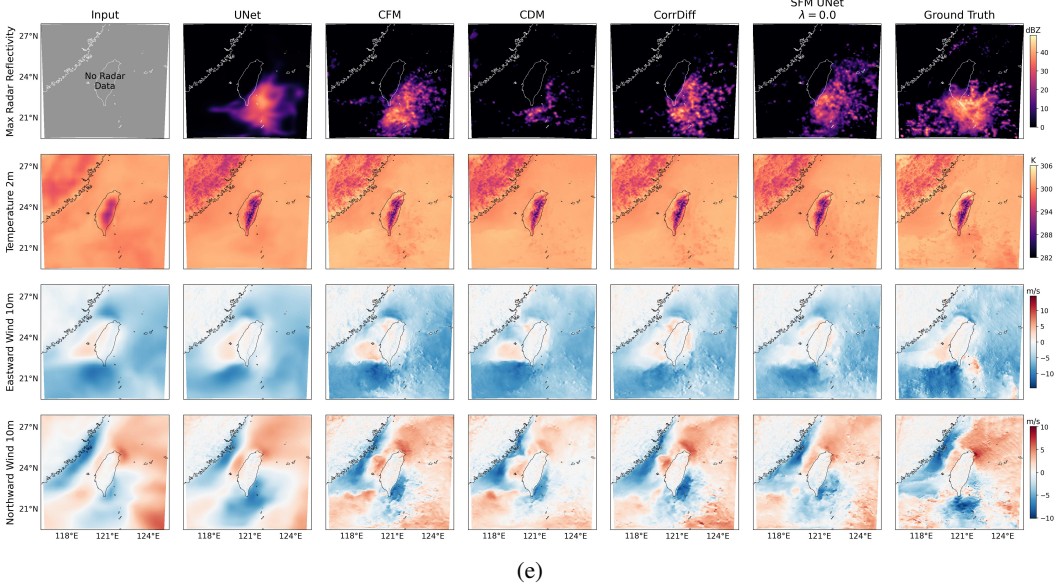

(e)

Figure 9: **(a-e) Visual Comparison of All Models for CWA Weather Data.** The AFM model demonstrates superior reconstruction quality, particularly in capturing fine-scale details, while other baselines show blurring or misalignment in key areas. (a-e) show the results for different models.

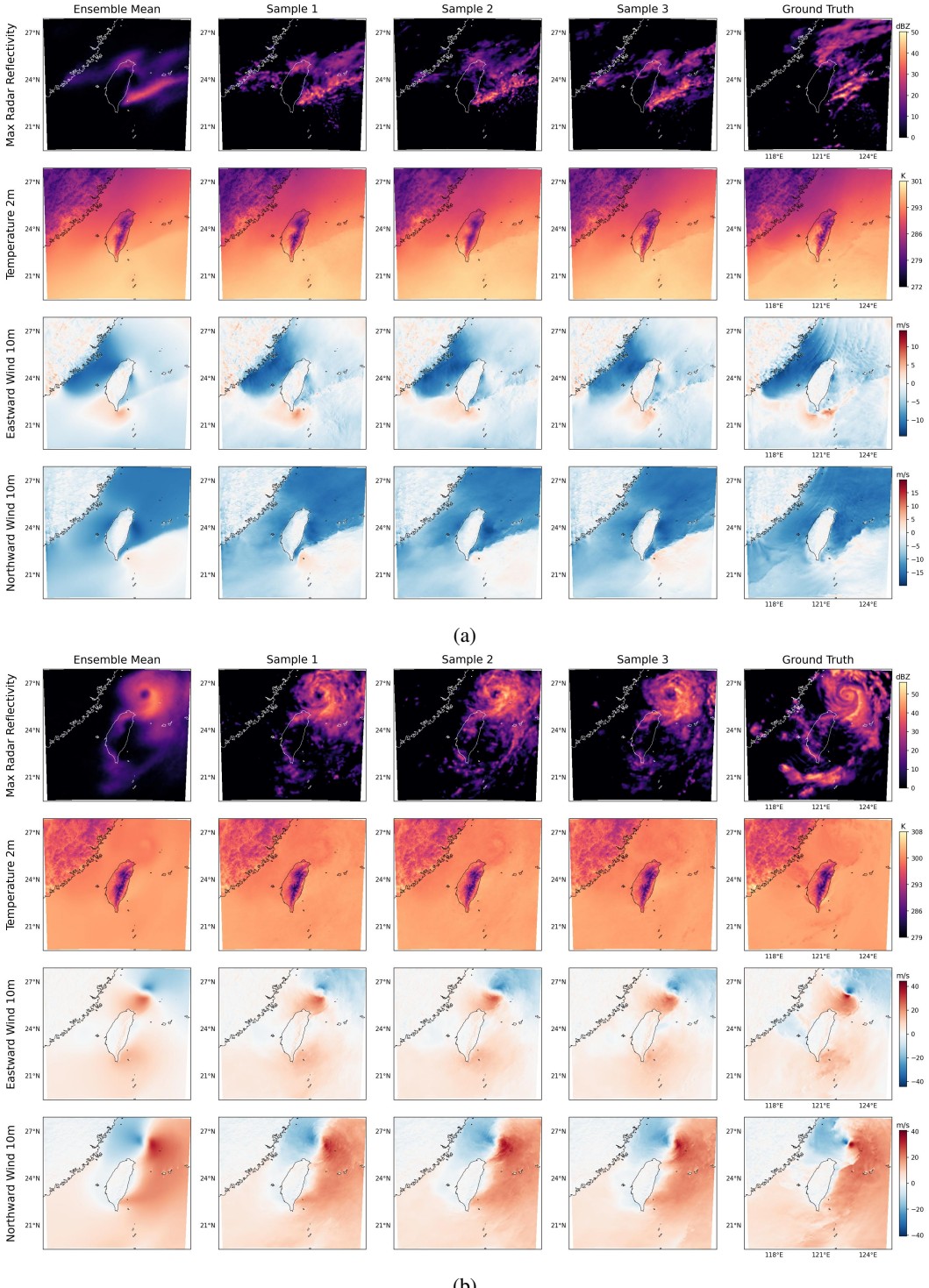

Figure 10: (Cont.)

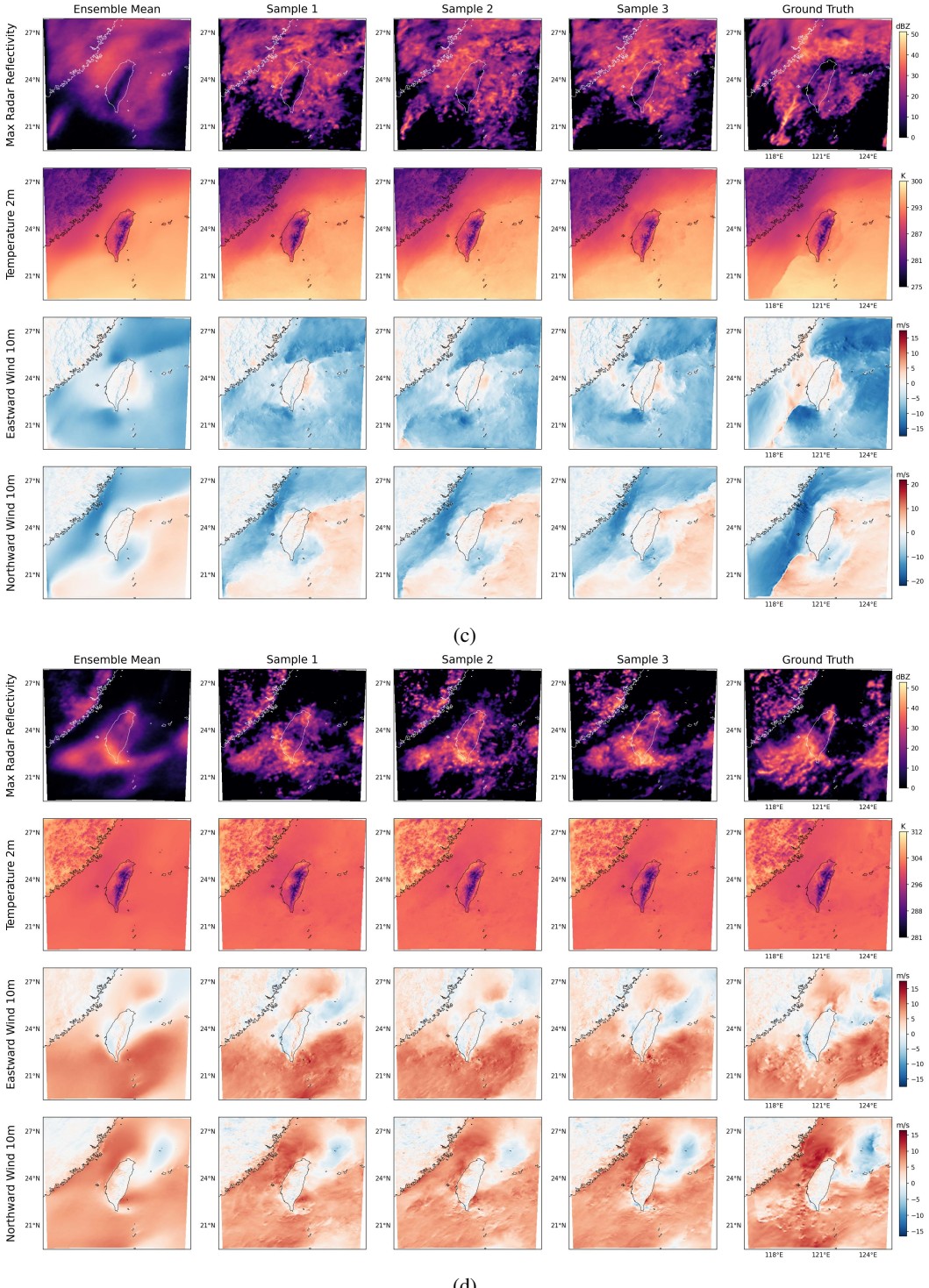

Figure 10: (Cont.)

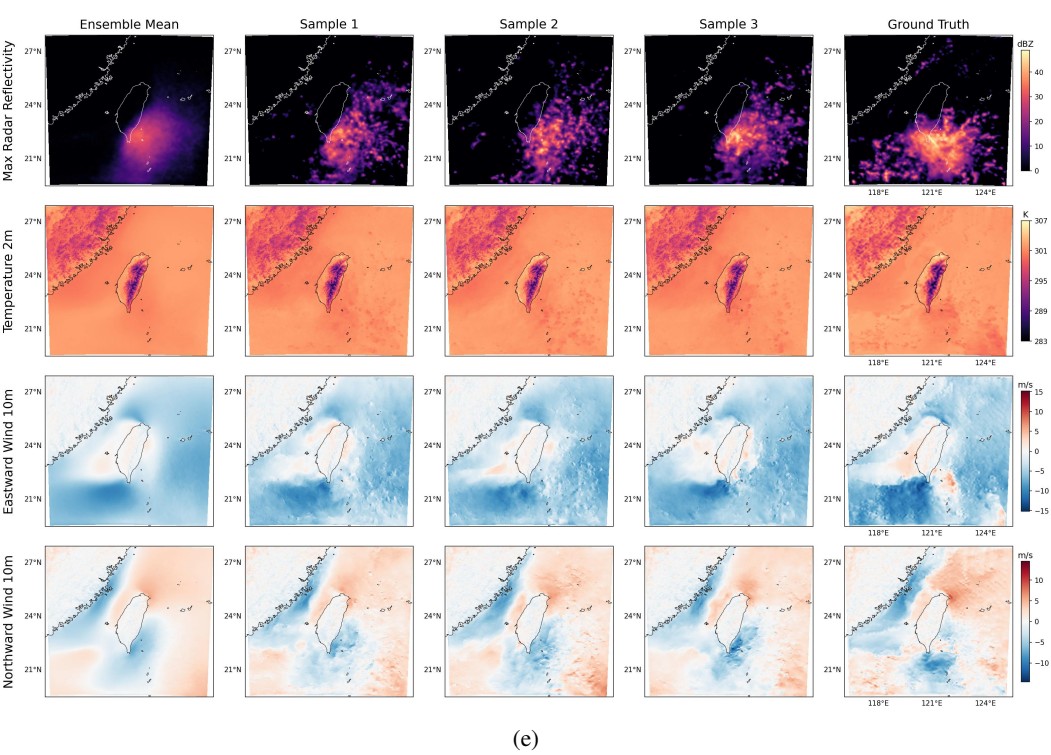

(e)

Figure 10: **Ensemble predictions for CWA Weather Data.** Results demonstrate AFM's ability to capture variable dynamics. (a-e) show the results for different points in time.

### B.5 KOLMOGOROV-FLOW DOWNSCALING

#### B.5.1 ABLATIONS

Different hyperparameters of both the dataset and the model are ablated, with the resulting metrics reported in Table 15. The results show that, in this scenario, where the data is relatively misaligned, a smaller encoder tends to achieve better generalization performance, possibly due to its capacity to focus on the most relevant features while avoiding overfitting. Additionally, the experiments demonstrate that conditioning the AFM with the coarse-resolution input enhances the predictive skill, highlighting the importance of incorporating multiscale information for improved downscaling accuracy. These findings provide valuable insights into the optimal model configurations for handling misaligned data.

| | Encoder | \multicolumn{4}{c}{$1 \times 1$conv} | \multicolumn{4}{c}{UNet} |
|---|---|---|---|---|---|---|---|---|---|
| | Adapt. $\sigma_z$ | ✗ | | ✓ | | ✗ | | ✓ | |
| $\tau$ | y cond. | ✗ | ✓ | ✗ | ✓ | ✗ | ✓ | ✗ | ✓ |
| **3** | **RMSE** ↓ | 1.22 | 0.91 | 1.22 | **0.73** | 1.15 | 1.11 | 1.17 | 1.17 |
| | **CRPS** ↓ | 0.63 | 0.48 | 0.62 | **0.37** | 0.65 | 0.63 | 0.69 | 0.70 |
| | **MAE** ↓ | 0.83 | 0.65 | 0.83 | **0.51** | 0.81 | 0.78 | 0.83 | 0.84 |
| | **SSR** → 1 | 0.56 | 0.58 | 0.58 | **0.62** | 0.37 | 0.34 | 0.31 | 0.30 |
| **5** | **RMSE** ↓ | 1.17 | 0.78 | 1.17 | **0.76** | 1.16 | 1.01 | 1.09 | 1.07 |
| | **CRPS** ↓ | 0.62 | 0.42 | 0.61 | **0.40** | 0.67 | 0.58 | 0.66 | 0.64 |
| | **MAE** ↓ | 0.82 | 0.56 | 0.82 | **0.54** | 0.83 | 0.72 | 0.80 | 0.78 |
| | **SSR** → 1 | 0.58 | 0.58 | **0.60** | 0.58 | 0.35 | 0.36 | 0.33 | 0.33 |
| **10** | **RMSE** ↓ | 1.36 | **1.06** | 1.39 | 1.09 | 1.35 | 1.36 | 1.36 | 1.28 |
| | **CRPS** ↓ | 0.71 | **0.57** | 0.78 | 0.65 | 0.79 | 0.79 | 0.80 | 0.74 |
| | **MAE** ↓ | 0.96 | 0.78 | 0.98 | **0.77** | 0.98 | 0.99 | 1.00 | 0.95 |
| | **SSR** → 1 | **0.69** | 0.63 | 0.43 | 0.23 | 0.36 | 0.37 | 0.38 | 0.45 |

Table 15: **Kolmogorov Flow Ablation Study for AFM:** This table examines the effect of different hyperparameters on performance across misalignment levels ($\tau$). A smaller encoder with conditioning consistently performs better for highly misaligned data. Additionally, adaptive noise scaling ($\sigma_z$) enhances performance when conditioning AFM on coarse-resolution input data (**y**).

#### B.5.2 ENSEMBLE ANALYSIS

Representative KF samples along with the generated ensemble members are depicted in Figs. 13a, 13b, and 13c. These figures illustrate the variability captured by the ensemble across different forecast lead times. The diversity in the ensemble members indicates the model's ability to represent the inherent uncertainty in the system. Furthermore, the alignment of the ensemble mean with the observed samples suggests that the model not only captures the central tendency but also effectively characterizes the stochastic nature of the dynamics.

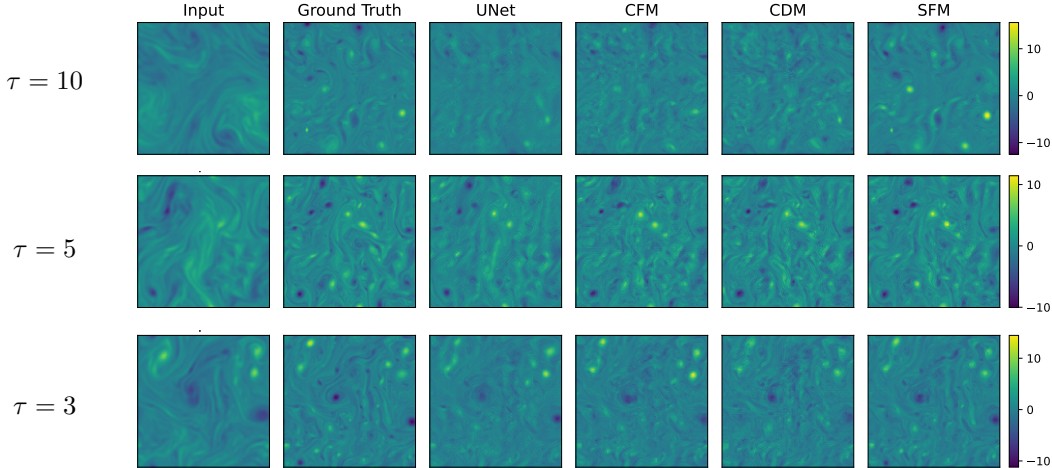

Figure 11: **AFM vs. Baselines for Different Misalignment Levels in Kolmogorov Flow Downscaling:** Each row corresponds to a different misalignment level $\tau$ (left side). From top to bottom, the rows represent $\tau = 10$, $\tau = 5$, and $\tau = 3$. As misalignment increases, the AFM significantly outperforms baseline models by generating samples that better align with the target distribution. Additionally, note the presence of high-frequency artifacts in the baseline models, which are more noticeable when the figures are zoomed in.

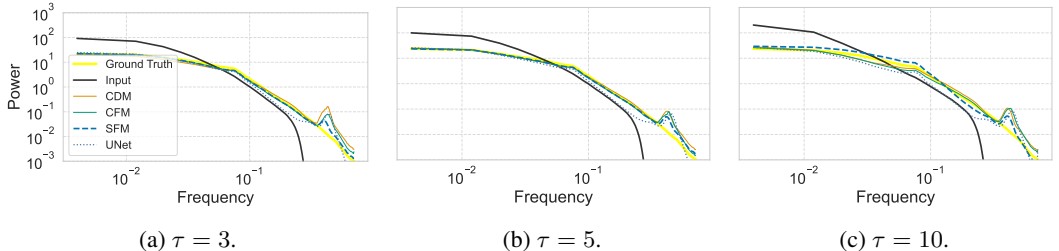

Figure 12: **AFM spectra vs. baselines for the Kolmogorov Flow.** AFM maintains superior fidelity to the ground truth across different $\tau$ values, highlighting its robustness in preserving both small and large-scale structures under various misalignment conditions. The small bump around the middle is caused by the energy that is added in the system (see appendix A.4.2).

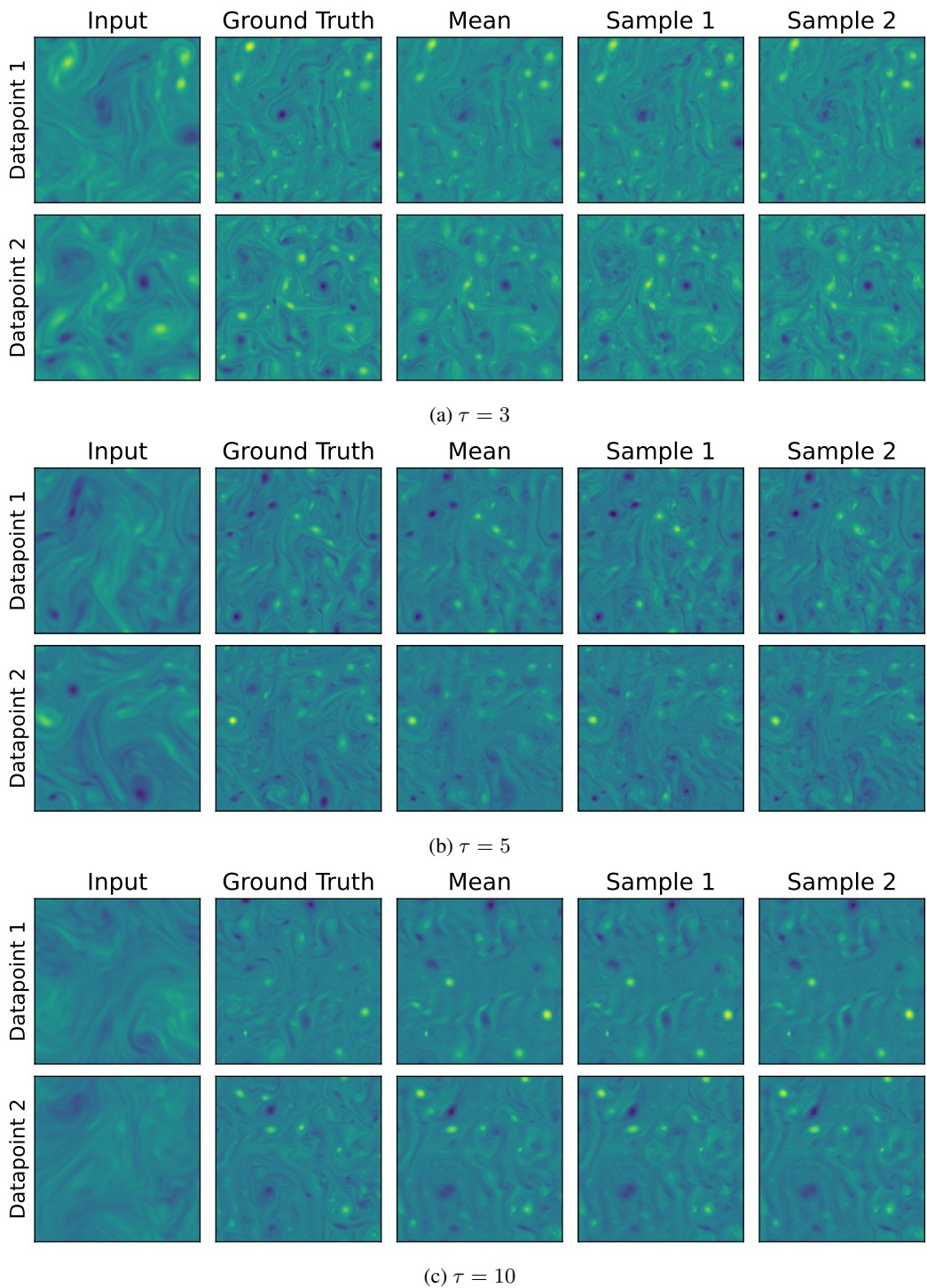

Figure 13: **Ensemble Predictions of AFM for Kolmogorov Flow at Different $\tau$ Values.** AFM ensemble predictions are shown for different $\tau$ values ($\tau = 3, 5, 10$), illustrating the model's ability to capture the variability and dynamics of the Kolmogorov flow across increasing levels of misalignment.

