# OpenReview forum: "Stochastic Flow Matching for Resolving Small-Scale Physics"
_ICLR.cc/2025/Conference — Submitted to ICLR 2025_

### Official Review · Reviewer_iuGH · 2024-10-28

**Soundness:** 3
**Presentation:** 3
**Contribution:** 3
**Rating:** 5
**Confidence:** 4

**Summary:**

The paper tackles a difficult variant of super-resolution applied to weather data where the goal is to retrieve small-scale dynamics of (potentially different) physics variables from a coarser observation grid. The authors build upon Mardani et al., 2023, proposing to disentangle the deterministic part (modeled with a U-Net or a 1x1 Conv.) and the stochastic part (modeled with flow matching). Conversely to Mardani et al., 2023, the authors jointly trained both parts by dynamically scaling the influence of the stochasticity in order to prevent overfitting.

**Strengths:**

The paper is well written (despite some minors typos) and easy to follow. The method is well explained and sounds reasonable. The authors conducted an impressive set of experiments, anticipating well potential questions of the reader.

**Weaknesses:**

**Major**
- The authors claims that their method is *specifically tailored for data-limited regimes*, and mention this as a key limitation of Mardani et al., 2023 compared to their approach. Yet, no experiment explore low data-regime. The ERA5-CWA dataset contains 4 years of hourly sampled data, and the "toy" dataset contains 100k points.
- I found section 4.2 / 4.3 rather unconvincing, especially the intuitive motivation for the adaptive noise scaling. I think these sections could benefit from a clarification from the authors (see related questions).

**Minor**
- In the introduction, the authors stated that the method is designed to match *spatially* misaligned data, yet in the rest of the paper, the term "misalignement" seems to refers to difference between the PDE governing the low-resolution data and the high-resolution one. It is not clear to me why such misalignment can be considered spatial. If *spatial* misalignement does corresponds to differences in PDEs, then the first paragraph of A.4.1 describes it very well and probably should be in the main paper.
- Replaying the proof of Proposition 1, when injecting the proposed form of $v_\theta(x,t) = (x_t - D_\theta(x_t, t)) / (1-t)$ into equation (7), I found:
$$\|v_\theta(x_t, t) - (x-z)\| = \left\|\frac{x_t - D_\theta(x_t, t)}{1-t} - (x-z)\right\|$$
then using equation (8):
$$\|v_\theta(x_t, t) - (x-z)\| = \left\|\frac{(1-t)z + tx - D_\theta(x_t, t)}{1-t} - (x-z)\right\| = \left\|2z + \frac{2tx - x}{1-t} - \frac{D_\theta(x_t,t)}{1-t}\right\|$$
So I think the correct expression for $v_\theta$ should rather be $v_\theta(x,t) = \frac{x_t - D_\theta(x_t, t)}{t-1}$

**Grade motivation**
My grade is motivated mainly by my difficulties to fully understand what the proposed method actually learns, and consequently how it compares with pure deterministic or pure stochastic baselines. I suspect degeneration of the model toward a fully stochastic method, which seems supported by ablations, but contradicts main results in table 2. The "data-limited regime" claims is also confusing. The math typo is certainly a simple sign mistake, but also plays a role.

**Typo**
- Missing parenthesis in score function after equation (5)
- Repeated $\theta$ in A.1 (expression of $v_\theta$)

**Questions:**

**Major**
- Which parts of the model support your claim that it is *specifically tailored for data-limited regimes* ? Is there an experiment in the paper exposing the limitation of Mardani et al, 2023 in this context ?
- During training, how is $\sigma_z$  computed (using training or validation error) ?
- Both noise scaling and encoder regularization are motivated by the need to counter overfitting. Is there any proof in the paper that vanilla methods does suffer from overfitting that I could have missed ?
- If I understand correctly, the adaptive noise scaling will increase the noise amplitude if the deterministic part has large error. Then could it leads to a degenerate solution where $\sigma_z$ remains very large and most of the work is done by the stochastic part ? This seems supported by several results in the paper:
	- The simpler 1x1 conv. performs often better than UNet, which is unexpected, since UNet is more powerful. Could it indicates that the model is actually using more the stochastic part, and learns to ignore the deterministic part ?
	- Using the encoder regularization should prevent the degenerate case. And indeed, Table 6 shows that the regularization is helpful mostly for UNet.
I think the paper could benefit from a study on the strength of $\sigma_z$, for instance by analyzing how corrupted is the output of the deterministic part (if the noise is of the same order of magnitude, then the model has degenerated ?). It would also be interesting (if available) to see the evolution of $\sigma_z$ during training of the model for both UNet and Conv 1x1.

**Minor**
- Can you please confirm the typo in the proof of proposition 1 ?
- Could you please elaborate on Sec.4 (Deterministic Dynamics) "it spatially aligns the base and target distribution *by matching their large-scale dynamics*" ? The sentence is unclear to me.
- The authors suggest that $\sigma_z$ prevents overfitting by increasing when the validation error increases. This intuition is only valid when $\sigma_z$  is computed from a validation loss, hence not online. Then, what is the intuition behind *online* method for scaling the noise ?
- If the temperature is deterministic, how the authors explains that the UNet Reg. baseline does not outperforms the other models on this variable ?

---

> ### Author Response · Authors · 2024-11-27
>
> Thanks for the positive and constructive comments. Having addressed your concerns one by one, we are looking forward to a higher grade. Please find below the response to your concerns:
>
> **Question 1:** The authors claim that their method is specifically tailored for data-limited regimes, and mention this as a key limitation of Mardani et al., 2023 compared to their approach. Yet, no experiment explores low data-regime. The ERA5-CWA dataset contains 4 years of hourly sampled data, and the "toy" dataset contains 100k points.
>
> **Response:** Apparently, there is a misunderstanding. Overfitting due to full regression training is only one of the challenges associated with CorrDiff that motivates AFM. AFM provides a principled way to formulate the superresolution of “misaligned” paired data in an end-to-end and adaptive fashion. Also, note that the argument about ERA5-CWA being considered a small data regime is valid since it only contains 35,000 samples for 4 years of hourly data acquisition.
>
> **Question 2:** I found section 4.2 / 4.3 rather unconvincing, especially the intuitive motivation for the adaptive noise scaling. I think these sections could benefit from clarification from the authors (see related questions).
>
> **Response:** Thank you for your feedback. We have added clarifying statements to sections 4.2 and for improved clarity, added Section B.1 in the appendix to discuss the overfitting issue in more depth. We reply to your specific questions below. We would appreciate the reviewers comment to further clarify it.
>
> **Question 3:** In the introduction, the authors stated that the method is designed to match spatially misaligned data, yet in the rest of the paper, the term "misalignment" seems to refer to the difference between the PDE governing the low-resolution data and the high-resolution one. It is not clear to me why such misalignment can be considered spatial. If spatial misalignment does correspond to differences in PDEs, then the first paragraph of A.4.1 describes it very well and probably should be in the main paper.
>
> **Response:** Thanks for your comment. We have used the PDE misalignment and spatial misalignment interchangeably. The idea is that misaligned PDEs lead to spatially misaligned data as the pixels can shift and deform spatially. For example in the CWA data, when a typhoon is present the eye of the typhoon is spatially misaligned between the low and high resolution simulations. This is the outcome of divergent simulations due to different dynamic models used in each scale. We clarify this in the introduction of the revised paper.
>
> **Question 3:** Replaying the proof of Proposition 1, when injecting the proposed form of …
>
> **Response:** Thank you very much for pointing out the oversight in the sign. We apologize for the error. The correct vector field is indeed:
> $$\boldsymbol{v}_{\theta}(\boldsymbol{x}_t, t) = \frac{D(\boldsymbol{x}_t, \sigma_t) - \boldsymbol{x}_t}{1-t}, $$
> which aligns with the reviewer’s derivation. We have updated the manuscript accordingly.

---

> ### Author Response · Authors · 2024-11-27
>
> Please find below our responses to the rest of your concerns:
>
> **Question 4:** My grade is motivated mainly by my difficulties to fully understand what the proposed method actually learns, and consequently how it compares with pure deterministic or pure stochastic baselines. I suspect degeneration of the model toward a fully stochastic method, which seems supported by ablations, but contradicts main results in table 2. The "data-limited regime" claims is also confusing. The math typo is certainly a simple sign mistake, but also plays a role.
>
> **Response:** Thanks for the explanation and giving us the chance to address your concerns. We believe we have fully addressed them, and we are looking for a more favorable grade. More concretely:
>
> - **Pure deterministic baseline ($\lambda \rightarrow \infty$):** the best way to understand the extreme cases is to look at the perturbation kernel in Algorithm 1 (line 6). If the encoder fully predicts the data. i.e., when $\boldsymbol{x} = \mathcal{E}(\boldsymbol{y})$, then $e = 0$, and thus $\sigma_z = \sigma_t = 0, \forall t$, which leads to **no** stochasticity in the forward generative process.
> - **Pure stochastic baseline ($\lambda \rightarrow 0$):** again, considering the perturbation kernel in Algorithm 1 (line 6), if the regression error is large, then $\sigma_z = \|e\|$ tends to be large and thus $\sigma_t$ takes large values that lead to large variance for the base noise distribution. This obviously leads to aggressive exploration of the distribution via the generative process.
> - **Limited data regime:** there seems to be a misunderstanding here. A data limited regime is not the main motivation for this work. It's just a niche case that promotes overfitting. We provide additional experiments to show the train vs validation MSE divergence of UNet regression in Fig. 7 in the appendix of the revised manuscript. That clearly shows the overfitting issue when there are only (limited) 35k train samples. Please also see the response to question 1.
> - **Typos:** they are all fixed throughout the manuscript.
>
>
> **Question 5:** Missing parentheses in score function after equation (5)
>
> **Response:** Thank you very much for noticing this typo error. We have corrected it in the revised manuscript.
>
> **Question 6:** Repeated \theat in A.1 (expression of $u_{\theta}$)
>
> **Response:** Thank you very much for noticing this typo error. We have corrected it in the revised manuscript.
>
> **Question 7:** Which parts of the model support your claim that it is specifically tailored for data-limited regimes? Is there an experiment in the paper exposing the limitation of Mardani et al, 2023 in this context?
>
> **Response:** Data limited regime only increases the chance of overfitting. We provide additional evidence in the revised manuscript to confirm that UNet regression overfits; see Fig. 7. see also response to question 2 of reviewer ef9p. Note also that the CorrDiff results reported in Table 2 use early stopping already.
>
> **Question 8:** During training, how is $\sigma_z$ computed (using training or validation error)?
>
> **Response:** Great point. It is computed on the validation data. This indeed helps avoiding overfitting issues by dynamic adjustment. We clarify this in the revised manuscript.
>
> **Question 9:** Both noise scaling and encoder regularization are motivated by the need to counter overfitting. Is there any proof in the paper that vanilla methods do suffer from overfitting that I could have missed?
>
> **Response:** We provide additional evidence in the revised manuscript that pure UNet regression training overfits. Please see Fig. 7 that shows the train and validation MSE diverge quickly after 1-2M imgs and Table 5 for the effect on CorrDiff performance. We also copied Table 5 above in our response to reviewer AsTp.
>
> **Question 10:** If I understand correctly, the adaptive noise scaling will increase the noise amplitude if the deterministic part has large error. Then could it leads to a degenerate solution where $\sigma_z$ remains very large and most of the work is done by the stochastic part? This seems supported by several results in the paper:
>
> **Response:** Great observation. Yes, as seen in Algorithm 1 (line 6), if the data is very stochastic, the deterministic encoder leads to a large error, and thus $\sigma_z = \|x - \text{Enc}(y)\|^2$ is too large. As a result $\sigma_t$ is large, and thus the variance of base noise distribution grows significantly, which puts the heavy lifting on the generation side.

---

> > ### Author Response · Authors · 2024-11-27
> >
> > Please find below our responses to the rest of your concerns:
> >
> > **Question 11:** The simpler 1x1 conv. performs often better than UNet, which is unexpected, since UNet is more powerful. Could it indicate that the model is actually using more the stochastic part, and learns to ignore the deterministic part?
> >
> > **Response:** We hypothesize that UNet overfits the data and using a simple 1x1 conv acts as a regularization.
> >
> > **Question 12:** Using the encoder regularization should prevent the degenerate case. And indeed, Table 6 shows that the regularization is helpful mostly for UNet. I think the paper could benefit from a study on the strength of $\sigma_z$, for instance by analyzing how corrupted is the output of the deterministic part (if the noise is of the same order of magnitude, then the model has degenerated?). It would also be interesting (if available) to see the evolution of $\sigma_z$ during training of the model for both UNet and Conv 1x1.
> >
> > **Response:** This is a good point. In light of the reviewer suggestion, we ran new experiments to plot the evolution of $\sigma_z$ over the training steps for both UNet and 1x1 conv. Please see section xx in the revised manuscript. Please also see the response to question 7 of reviewer Bz1Z.
> >
> > **Question 13:** Can you please confirm the typo in the proof of proposition 1?
> >
> > **Response:** Thank you very much for noticing this typo error. We have corrected it in the revised manuscript.
> >
> > **Question 14:** Could you please elaborate on Sec.4 (Deterministic Dynamics) "it spatially aligns the base and target distribution by matching their large-scale dynamics"? The sentence is unclear to me.
> >
> > **Response:** To avoid confusion we rephrase the sentence as: “it first matches the large-scale, mainly deterministic dynamics of the input and output, aligning the spatially misaligned large-structures due to diverging trajectories”
> >
> > **Question 15:** The authors suggest that $\sigma_z$ prevents overfitting by increasing when the validation error increases. This intuition is only valid when $\sigma_z$ is computed from a validation loss, hence not online. Then, what is the intuition behind online method for scaling the noise?
> >
> > **Response:** We respectfully disagree. The main purpose of learning $\sigma_z$ online is computational efficiency by leveraging stochastic approximation following the recipe of online learning algorithms. Alternatively, we could cross validate “offline” for the best $\sigma_z$ that minimizes the validation reward. However, this requires a costly grid search to optimize $\sigma_z$ “per channel”.
> >
> > Overall, for online learning, “validation reward” is a legitimate choice, and quite aligned with online algorithms that take feedback from true future observations to correct their mistakes.
> >
> > **Question 16:** If the temperature is deterministic, how do the authors explain that the UNet Reg. baseline does not outperform the other models on this variable?
> >
> > **Response:** Good observation. UNet regression underperforms because it's fully trained and thus it overfits the data which leads to larger generalization error. To further clarify this comment, we have added train/validation loss for the UNet regression training in Fig. 7 of the revised manuscript that clearly shows the overfitting issue.

---

> > > ### Author Response · Authors · 2024-12-02
> > >
> > > Dear Reviewer iuGH,
> > > Since the deadline for reviewers posting for the authors is coming soon, It would be great if you could acknowledge our response and let us know if further clarifications on any of your questions is required. Thanks for your help in the review process.
> > > Best,
> > > Authors

---

> > > > ### Author Response · Authors · 2024-12-03
> > > >
> > > > Dear reviewer iuGH,
> > > >
> > > > As the review period ends we would like to summarize our efforts to address your main concerns:
> > > >
> > > > - **Evidence of Overfitting in Corrdiff**: Provided new experimental evidence (Section B.1) showing encoder overfitting in CorrDiff and detailed how our method balances deterministic and stochastic components to avoid degeneration into a fully stochastic or fully deterministic model.
> > > >
> > > > - **Improved analysis of adaptive scaling**: Added clarifications to sections 4.2 & 4.3 and added a new Section B.1 to provide deeper insights into the adaptive scaling dynamics during training.
> > > >
> > > > - **Explanation of Spatial Misalignment**: Clarified how misaligned PDEs can result in spatially misaligned data due to divergent simulations, updating the introduction for better understanding.
> > > >
> > > > - **Correction of Proof Error**: Fixed the sign error in Proposition 1's proof and added further clarifications to the proof.
> > > >
> > > > It would be great if you could acknowledge our response and let us know if further clarifications on any of your questions are required. Thanks for your help in the review process.
> > > >
> > > > Best,
> > > > Authors

---

### Official Review · Reviewer_ZwfG · 2024-10-31

**Soundness:** 3
**Presentation:** 3
**Contribution:** 3
**Rating:** 6
**Confidence:** 2

**Summary:**

The paper proposes stochastic flow matching for super-resolving small-scale physics in weather data, addressing challenges like data misalignment, multiscale dynamics, and limited data availability. The method combines an encoder that maps coarse-resolution inputs to a latent space with flow matching to generate fine-resolution outputs, while using adaptive noise scaling to balance deterministic and stochastic components. Experiments demonstrate that SFM outperforms existing methods like conditional diffusion and flow models.

**Strengths:**

- The presentation and writing are good

- The paper presents a novel combination of deterministic encoding with stochastic flow matching for solving the task

- The method's ability to handle misaligned data and limited data scenarios makes it valuable for certain applications

- Experiments on synthetic and real-world datasets demonstrate the effectiveness of the method

**Weaknesses:**

A theoretical analysis of SFM's convergence properties would be interesting, especially regarding the impact of the adaptive noise scaling mechanism on convergence guarantees

**Questions:**

- Could you provide more details about the hyperparameter selection process?

- Have you identified specific failure cases or limitations of the method?

---

> ### Author Response · Authors · 2024-11-26
>
> Thanks for the positive opinion and acceptance recommendation. Please find below the response to your concerns:
>
>
> **Question 1:** A theoretical analysis of SFM's convergence properties would be interesting, especially regarding the impact of the adaptive noise scaling mechanism on convergence guarantees.
>
> **Response:** Thank you for the insightful suggestion. A theoretical analysis of convergence properties, particularly regarding the adaptive noise scaling mechanism, is indeed important. However, this analysis is beyond the scope of the current manuscript. We plan to explore these theoretical aspects in future work and have acknowledged this limitation in the conclusion section of the revised manuscript.
>
> ---
> **Question 2:** Could you provide more details about the hyperparameter selection process?
>
> **Response:** We mainly need to tune $\lambda$ that controls the balance between deterministic and stochastic components. We use "offline" cross-validation with the validation set to optimize $\lambda$ for the best validation RMSE.
>
> ---
> **Question 3:** Have you identified specific failure cases or limitations of the method?
>
> **Response:** We have identified that the skill of our method diminishes as the degree of input-output misalignment increases. This limitation is demonstrated in our Kolmogorov flow experiments, where higher values of $\tau$ lead to lower skills (see Tables 3 and 14). However, even with increased misalignment, our method consistently outperforms baseline models. Additionally, this degradation in performance with higher $\tau$ values is visually evident in Figure 10 and further corroborated by the spectral analyses presented in Figure 11.
>
> ---

---

> > ### Author Response · Authors · 2024-12-03
> >
> > Dear reviewer ZwfG,
> >
> > As the review period ends we would like to summarize our efforts to address your main concerns:
> >
> > - **Hyperparameter Selection Details**: Provided more information on how we tune $\lambda$ using cross-validation to optimize RMSE on the validation set.
> >
> > - **Limitations and future work**: Added remarks to the conclusions section, noting the decreases performance with higher input-output misalignment (as shown in the Kolmogorov flow experiments), but still outperforms baseline models. Recognized the importance of a theoretical convergence analysis of AFM and plan to pursue this in future work.
> >
> > - **Analysis of adaptive noise scaling during training**: We added Section B.2 (incl. Fig. 8) in the revised manuscript that discusses the evolution of $\sigma_z$ over training steps for various channels (i.e. variables).
> >
> > - **Evidence of Overfitting in CorrDiff**: Added new experimental results in Section B.1 demonstrating overfitting issues in CorrDiff and how they greatly affect ensemble variance.
> >
> > It would be great if you could acknowledge our response and let us know if further clarifications on any of your questions are required. Thanks for your help in the review process.
> >
> > Best,
> > Authors

---

### Official Review · Reviewer_ef9p · 2024-11-03

**Soundness:** 3
**Presentation:** 3
**Contribution:** 2
**Rating:** 5
**Confidence:** 3

**Summary:**

This paper presents a framework that becomes stochastic stream matching for learning to recover small-scale physical fields. Specifically, they first use an encoder to learn to regress deterministic dynamics, and then add stream matching to learn the distribution of fine-grained stochastic dynamics. They validate the algorithm on multi-scale Kolmogorov flow and regional weather downscaling datasets.

**Strengths:**

1. The presentation is apparent and the description of the proposed methodology is accurate and easy to follow.

2. The figure and quantitative comparisons are good, the experiment is sufficiently adequate, the experimental setup is described in detail with methodological details and it should be easily reproducible.

**Weaknesses:**

1. The problem of novelty, due to the well-known connection between flow matching and diffusion modeling. The approach proposed in this paper seems to resemble CorrDiff. Although the authors emphasize the difference in Section 4.4. They claim that CorrDiff is trained in two stages, i.e., the regression encoder is trained first and the diffusion of the residual components is trained afterwards. In contrast, it seems that in this paper, the two components are only trained jointly, and the final loss is the sum of these two components.

2. In addition, the authors claim that the first stage of CorrDiff may run the risk of overfitting. This reasoning seems insufficient to me, and the authors don't seem to have relevant evidence. There is also no guarantee that joint training avoids this risk, besides this can be solved perfectly well using simpler ways, such as early stopping.

3. The idea of co-training encoders and diffusion models is also not new, and in fact he has already proposed it for tasks such as speech synthesis [1,2] and precipitation nowcasting [3].

[1] Popov V, Vovk I, Gogoryan V, et al. Grad-tts: A diffusion probabilistic model for text-to-speech[C]//International Conference on Machine Learning. PMLR, 2021: 8599-8608.

[2] Chen Z, He G, Zheng K, et al. Bridge-TTS: Text-to-Speech Synthesis with Schrodinger Bridge[J].


[3] DiffCast: A Unified Framework via Residual Diffusion for Precipitation Nowcasting

**Questions:**

see Weaknesses

---

> ### Author Response · Authors · 2024-11-27
>
> Thank you for the positive feedback and constructive comments. Please find below the response to your concerns:
>
> **Question 1:** The problem of novelty, due to the well-known connection between flow matching and diffusion modeling. The approach proposed in this paper seems to resemble CorrDiff. Although the authors emphasize the difference in Section 4.4. They claim that CorrDiff is trained in two stages, i.e., the regression encoder is trained first and the diffusion of the residual components is trained afterwards. In contrast, it seems that in this paper, the two components are only trained jointly, and the final loss is the sum of these two components.
>
> **Response:** We respectfully disagree. CorrDiff is a two-stage method. When the regression model overfits in the first stage, the second-stage diffusion model generalizes poorly since it mostly relies on the residual data obtained from the regression model. Our approach fixes this issue with two significant modifications:
>
> - The weight $\lambda$ in the joint-loss controls the balance between deterministic and stochastic components. This allows integrating physical knowledge of the variables through weighting the loss, thus balancing the degree of deterministic behavior.
> - We added the “adaptive” noise scaling mechanism based on the maximum likelihood criterion, which informs the generative component about the level of stochasticity. This makes the generative component “robust” to regression errors.
>
> For instance, our experiments in Tables 6 and 11 show that for temperature, which is a highly deterministic variable, there is a middle range for lambda that yields the best results. This is more prevalent with UNet (see Table 11), indicating that putting more emphasis on the encoder increases overfitting, similar to CorrDiff.
>
> **Question 2:** In addition, the authors claim that the first stage of CorrDiff may run the risk of overfitting. This reasoning seems insufficient to me, and the authors don't seem to have relevant evidence. There is also no guarantee that joint training avoids this risk, besides this can be solved perfectly well using simpler ways, such as early stopping.
>
> **Response:** Thank you for your insightful question, which helped us clarify this point in the revised manuscript. CorrDiff is indeed susceptible to overfitting because, during training, the supervised regression component can overfit the data, resulting in near-zero residuals. This means the diffusion model only encounters residuals concentrated around zero with little variation, leading it to generate negligible corrections. However, in the test phase, the residuals are significantly larger and out of distribution, rendering the diffusion model ineffective at correcting them.
> We added new experiments (see Fig 7.) demonstrating the overfitting issue with UNet regression, where the training and test losses diverge rapidly. Notably, the CorrDiff results reported in Table 2 already incorporate early stopping, yet early stopping alone does not adequately address the overfitting problem.
> While we agree that joint training can also risk overfitting, our AFM method includes two key mechanisms to mitigate this:
>
> - The $\lambda$ Coefficient: Balances the MSE loss, which is inherently prone to overfitting.
> - Adaptive Noise Scaling: Receives feedback from "validation" data to guide the generative model, ensuring it anticipates the true level of uncertainty during testing and remains within the appropriate distribution.
> To further address this concern, we have added a new subsection (Section B.1 of the Appendix) in the revised manuscript comparing AFM with CorrDiff in detail.
>
> **Question 3:** The idea of co-training encoders and diffusion models is also not new, and in fact he has already proposed it for tasks such as speech synthesis [1,2] and precipitation nowcasting [3].
>
> **Response:** We thank the reviewer for bringing these relevant references to our attention. We have now cited and discussed them in Section 2 of the revised manuscript. However, we would like to highlight the following distinctions between these methods and our work:
>
> - Different Tasks: DiffCast concentrates on temporal next-frame prediction, and the other two methods on text-to-speech synthesis. This is fundamentally different from our super-resolution task and channel reconstruction task.
> - Spatial Misalignment: The methods utilize a single scale between inputs and outputs, they do not face the spatial misalignment challenges that we address. Our method, AFM, handles multichannel generation with fully misaligned input and output.
> - Multiple Channels with Different Stochasticity: These methods operate on a single “channel”. AFM tackles multiple channels with different stochasticity characteristics at once.

---

> > ### Author Response · Authors · 2024-12-02
> >
> > Dear Reviewer ef9p,
> > Since the deadline for reviewers posting for the authors is coming soon, It would be great if you could acknowledge our response and let us know if further clarifications on any of your questions is required. Thanks for your help in the review process.
> > Best,
> > Authors

---

> > > ### Author Response · Authors · 2024-12-03
> > >
> > > Dear reviewer ef9p,
> > >
> > > As the review period ends we would like to summarize our efforts to address your main concerns:
> > >
> > > - **Distinction from CorrDiff**: Elaborated on the key differences, highlighting how our joint training approach with the $\lambda$ coefficient and adaptive noise scaling addresses overfitting and issues with data containing channels with varied stochasticity.
> > >
> > > - **Analysis of adaptive noise scaling during training**: We added Section B.2 (incl. Fig. 8) in the revised manuscript that discusses the evolution of $\sigma_z$ over training steps for various channels (i.e. variables).
> > >
> > > - **Evidence of Overfitting in CorrDiff**: Added new experimental results in Section B.1 demonstrating overfitting issues in CorrDiff and how they greatly affect ensemble variance.
> > >
> > > - **Related Work Discussion**: Acknowledged the co-training methods in other domains and discussed them in Section 2. We emphasized how our work differs in task focus, handling of spatial misalignment, and multi-channel stochasticity.
> > >
> > > It would be great if you could acknowledge our response and let us know if further clarifications on any of your questions are required. Thanks for your help in the review process.
> > >
> > > Best,
> > > Authors

---

### Official Review · Reviewer_AsTp · 2024-11-03

**Soundness:** 2
**Presentation:** 3
**Contribution:** 2
**Rating:** 5
**Confidence:** 3

**Summary:**

This paper presents a stochastic flow matching approach for super-resolving small-scale details. Specifically, it uses an encoder to transform inputs into latent images that are more aligned with the ground truth images. The additional noise is added to the latent images to introduce uncertainty for the following flow matching. The authors further present a modified flow matching objective that involves the noisy latent in training. Moreover, the maximum likelihood learning is proposed to tune the noise scale and an additional regularization is used in encoder learning for better generalization. The proposed method is evaluated on multi-scale weather datasets and synthetic PDE datasets.

**Strengths:**

1. The overall idea for cross-modality image translation is promising.
2. The proposed stochastic encoder-denoiser pipeline is interesting.
3. Extensive Experiments illustrate the effectiveness of the proposed method, and the results on most datasets are good.

**Weaknesses:**

1. If the authors want to add uncertainty to the deterministic part, maybe the better way is to apply approaches like VAE to output both mean and variance (the downscale layers can be removed if there is no resolution change). Using VAE can also avoid the tuning of noise scales.
2. Also, calling this method "stochastic flow matching" is somehow improper, since the flow matching is usually an ODE. Note that the original flow matching also accepts noise input.   Adding the "stochastic" to "flow matching" makes it confusing and readers might think it's a general method converting flow matching to an SDE.
3. In Proposition 1, how do you define the velocity field $v$? Moreover, I don't think the term $(1-t)$ can be simply ignored since it changes over time. At least the authors should add some experiments to show that ignoring the $1-t$ could achieve the same performance.
4. As stated in Sec 4.2, the noise level is highly related to x and y and thus this work proposes a maximum likelihood approach to tune it. My question is why not learn it through the encoder directly? It should be easier and can be trained in an end-to-end manner as it is also involved in the training objective.
5. Adding the encoder regularization is too intuitive and would definitely affect the learning of flow matching. More details and ablation experiments should be provided.
6. The 1 x 1 conv layer encoder performs surprisingly well compared to the complex UNet encoder. Can the authors provide more analysis on it? Why a linear transform can lead to better performance?
7. In appendix A.1, the last paragraph, it should be D_{\theta}.

**Questions:**

See "Weaknesses".

---

> ### Author Response · Authors · 2024-11-27
>
> Thank you very much for the positive feedback and constructive comments. Please find below the response to your concerns:
>
> **Question 1:** If the authors want to add uncertainty to the deterministic part, maybe the better way is to apply approaches like VAE to output both mean and variance (the downscale layers can be removed if there is no resolution change). Using VAE can also avoid the tuning of noise scales.
>
> **Response:** This is a valid point. Indeed, we could alternatively impose a KL regularization on the encoder output to predict both $\mu$ and $\sigma$ using the encoder, and use the dispersion $\sigma$ to drive the noise for FM. However, instead, we opted for a simpler formulation where we seek an encoder (i.e., $\mu$ and $\sigma$) that maximizes the likelihood of output. This is a well-studied problem also in VAEs (see [1]) with a closed-form solution for $\sigma$ as discussed in the paper. We found this simplicity elegant and in line with the domain-specific intuition that a deterministic predictor can to some good degree predict the output given input.
>
> Note that although VAE-based formulation would allow us to learn $\sigma$ automatically, it would require tuning the KL regularization and it can suffer from the prior hole problem. Having said that we agree that that is a viable approach and worth pursuing as the future research work. We add a remark to the revised manuscript to discuss this point (see the remark at the end of Section 4.2). We also  will highlight this as a viable future work in the conclusions section.
>
>
> [1] Simple and Effective VAE Training with Calibrated Decoders, Rybkin et al., ICLR 2021
>
> **Question 2:** Also, calling this method "stochastic flow matching" is somehow improper, since the flow matching is usually an ODE. Note that the original flow matching also accepts noise input. Adding the "stochastic" to "flow matching" makes it confusing and readers might think it's a general method converting flow matching to an SDE.
>
> **Response:** This is a valid point. It’s worth noting that flow matching also has stochasticity injected in the base distribution which is often assumed to be a standard Normal distribution. In our framework, the noise injected in the output of the encoder plays the same role and turns our model into a generative model.
>
> All said, to avoid any confusion around our proposal, we decided to change the name to “adaptive flow matching”, where the noise is “adapted” to the encoder error.
>
> **Question 3:** In Proposition 1, how do you define the velocity field u? Moreover, I don't think the term $(1-t)$  can be simply ignored since it changes over time. At least the authors should add some experiments to show that ignoring the (1-t) could achieve the same performance.
>
> **Response:** Thank you for your insightful comment, which allowed us to clarify our proof. We do not omit the $(1-t)$ term. Instead, we define $\sigma_t = (1-t) \sigma_z$, allowing us to rewrite the expression as $1/(1-t) =  \sigma_z / \sigma_t$. To enhance clarity, we have expanded the proof of Proposition 1 in the appendix. Additionally, we have included the reparameterization of the vector field as a denoiser (note that in Flow Matching, time is reversed relative to diffusion models, progressing from 0 to 1): $$\boldsymbol{v}_{\theta}(\boldsymbol{x}_t, t) = \frac{ D(\boldsymbol{x}_t, \sigma_t) - \boldsymbol{x}_t }{1-t}$$
>
> **Question 4:** As stated in Sec 4.2, the noise level is highly related to x and y and thus this work proposes a maximum likelihood approach to tune it. My question is why not learn it through the encoder directly? It should be easier and can be trained in an end-to-end manner as it is also involved in the training objective.
>
> **Response:** Your point related to Question 1 is valid. While VAE-based approaches that jointly learn $(\mu, \sigma)$ are viable solutions, our physics-inspired method offers an initial step toward addressing this problem. Also, note that the closed form $\sigma_z$ allows us to have a robust estimation using a validation set. This way if the encoder overfits, RMSE on the validation set will be higher and thus we will inject more noise in the output of the encoder adaptively, making the flow-matching model robust to the potential error made by the encoder.

---

> ### Author Response · Authors · 2024-11-27
>
> Please find below our responses to the rest of your concerns:
>
> **Question 5:** Adding the encoder regularization is too intuitive and would definitely affect the learning of flow matching. More details and ablation experiments should be provided.
>
> **Response:** Encoder regularization is crucial for balancing the deterministic and stochastic components of predictions. Our ablation studies (see Tables 6 and 11) show that lower $\lambda$ provide best results for radar which is the most stochastic variable. On the other hand, for highly deterministic variables like temperature, moderate $\lambda$ is the best. This seems to suggest that higher $\lambda$ can lead to overfitting and is more prevalent when using a UNet encoder. Furthermore, lower \lambda  allows increased ensemble variability, as indicated  by SSR, and for $\lambda=0$ SFM acquires the best variability over all models. These findings confirm that appropriate encoder regularization enhances the generalization of flow matching in our framework. We have added further discussion in Section B.1.1 of the revised manuscript.
>
> **Question 6:** The 1 x 1 conv layer encoder performs surprisingly well compared to the complex UNet encoder. Can the authors provide more analysis on it? Why a linear transform can lead to better performance?
>
> **Response:** Good observation. We found that the UNet encoder tends to overfit the training data, especially in data-limited settings. This behavior was also observed in the CorrDiff paper. This overfitting reduces the stochasticity necessary for effective flow matching, leading to large generalization errors during testing. In contrast, the simpler 1×1 convolutional encoder avoids overfitting and maintains the variability needed for flow matching to correct errors effectively. This balance between simplicity and variability allows the linear convolution to generalize better, resulting in superior performance compared to the more complex UNet encoder.
>
> **Question 7:** In appendix A.1, the last paragraph, it should be $D_{\theta}$.
>
> **Response:** Thank you very much for noticing this typo error. We have corrected it in the revised manuscript.

---

> ### Author Response · Authors · 2024-11-28
> **Table: Impact of Disjoint Regression Training on CorrDiff**
>
> We copy here Table 5 from the revised manuscript. It shows the effect of UNet training steps on CorrDiff performance.
> More results on the UNet's ovefitting issue can be found in Fig. 7.
>
> **Impact of Disjoint Regression Training on CorrDiff**
>
> This table demonstrates how varying the number of training steps for the UNet regression in the first stage affects the final performance of CorrDiff. UNet checkpoints trained for 0.5M, 2M, and 50M steps were utilized to train different diffusion models in the second stage. The results show that less-trained UNet models are better calibrated and exhibit superior stochasticity in their outcomes (SSR → 1). In contrast, increased training leads to less diverse ensembles, suggesting that the model struggles to correct a biased UNet while maintaining variability in its results.
>
> | **Variable**       | **Metric**        | **0.5M** | **2M** | **50M** |
> |--------------------|-------------------|----------|--------|---------|
> | **Radar**          | **RMSE ↓**        | 5.08     | 5.28   | 5.13    |
> |                    | **CRPS ↓**        | 1.81     | 1.89   | 2.10    |
> |                    | **SSR → 1**       | 0.52     | 0.35   | 0.14    |
> | **Eastward Wind**  | **RMSE ↓**        | 1.51     | 1.53   | 1.48    |
> |                    | **CRPS ↓**        | 0.83     | 0.89   | 0.87    |
> |                    | **SSR → 1**       | 0.60     | 0.42   | 0.39    |
> | **Northward Wind** | **RMSE ↓**        | 1.72     | 1.70   | 1.65    |
> |                    | **CRPS ↓**        | 0.97     | 1.01   | 0.99    |
> |                    | **SSR → 1**       | 0.56     | 0.39   | 0.37    |
> | **Temperature**    | **RMSE ↓**        | 0.96     | 0.93   | 0.91    |
> |                    | **CRPS ↓**        | 0.57     | 0.58   | 0.53    |
> |                    | **SSR → 1**       | 0.41     | 0.28   | 0.31    |

---

> > ### Author Response · Authors · 2024-12-02
> >
> > Dear Reviewer AsTp,
> > Since the deadline for reviewers posting for the authors is coming soon, It would be great if you could acknowledge our response and let us know if further clarifications on any of your questions is required. Thanks for your help in the review process.
> > Best,
> > Authors

---

> ### Author Response · Authors · 2024-12-03
>
> Dear reviewer AsTp,
>
> As the review period ends we would like to summarize our efforts to address your main concerns:
>
> - **Discussion on Learning \(\sigma_z\)**: While we opted for a simpler, closed-form solution for \(\sigma_z\), we acknowledge that learning \(\sigma\) through the encoder (e.g., using VAE approaches) is viable and have added a remark discussing this potential extension.
>
> - **Clarification on Velocity Field**: Provided a clearer definition of the velocity field in Proposition 1 to show how the (1-t) factor is accounted for.
>
> - **Analysis of Encoder Overfitting**: Included detailed ablation experiments (Tables 6 and 11) and expanded discussion in Section B.1.1 to show how encoder overfitting affects performance.
>
> - **Method Name**: Renamed our approach to "Adaptive Flow Matching" to avoid confusion with established concepts.
>
> - We have also corrected typos and other minor errors.
>
> It would be great if you could acknowledge our response and let us know if further clarifications on any of your questions is required. Thanks for your help in the review process.
>
> Best,
> Authors

---

### Official Review · Reviewer_Bz1Z · 2024-11-04

**Soundness:** 2
**Presentation:** 3
**Contribution:** 2
**Rating:** 5
**Confidence:** 3

**Summary:**

The paper introduces the Stochastic Flow Matching (SFM) framework, a novel approach to super-resolve small-scale physics in meteorological data, particularly in downscaling weather variables for regions covering Taiwan. The authors address three key challenges: spatial misalignment between input and output distributions, deterministic vs. stochastic dynamics across scales, and limited data, which can lead to overfitting. SFM leverages an encoder to map coarse-resolution inputs to a latent distribution aligned with the target fine-resolution, which is refined using flow matching to reconstruct small-scale stochastic details. An adaptive noise scaling mechanism, based on maximum-likelihood estimates, helps balance deterministic and stochastic components. The framework is tested on real-world CWA weather data and the PDE-based Kolmogorov dataset, consistently outperforming existing methods, such as conditional diffusion and flows, in terms of RMSE and fidelity across several variables and spatial resolutions.

**Strengths:**

- The authors conducted comprehensive experiments on both synthetic and real-world datasets.

- The paper includes ablation studies and spectral analysis, which reveal the impact of different architectural choices and underscore SFM's robustness in capturing high-frequency details.

- The dynamic adjustment of noise levels based on the encoder’s prediction error provides a robust method to balance deterministic and stochastic data components.

**Weaknesses:**

- It is not clear why the method is named and formalized as "flow matching" because the actual implementation is based on EDM. The perturbation and the denoising model both follow the diffusion custom instead of flow matching. Please justify whether the method especially the probability path is closer to flow matching or diffusion.

- In Algorithm 2. the inputs to the denoising model $\mathcal{D}_\theta$ are $(x, t)$ but elsewhere $(x, \sigma)$. This may lead to confusion on how the adaptive noise scaling can affect sampling.

-  Application of the proposed method to other domains is unexplored, limiting the generalizability of the findings. It seems that the method can be applied to general image-to-image translation tasks, for example restoration of natural images which also involves stochasticity in generation of local details.

- As acknowledged in the paper, SFM relies on paired input-output data, which may not always be feasible in real-world applications, particularly for unpaired or missing datasets.

- Improvement upon baseline method CorrDiff is not consistent (especially temperature in Table 2).

- A minor point that does not affect the score (if the method still modelled as flow matching).  SFM/SfM is commonly used in the broad field of AI to refer to Structure from Motion, which could lead to confusion, particularly if the paper is targeted at a broader audience that includes computer vision researchers.

**Questions:**

- See weaknesses 1 and 2.

- I wonder how the noise variance in the adaptive noise scaling varies in the training run, and if the effectiveness is sensitive to the hyperparameter setup.

**Details Of Ethics Concerns:**

The real-world weather data is restricted to the region covering the Taiwan island. I am uncertain if this would potentially lead to the bias towards over-tuning on tropical / subtropical regions, or raise other geographic or geopolical concerns.

---

> ### Author Response · Authors · 2024-11-26
> **Authors Response to Reviewers Comments**
>
> ---
> **Thanks for the positive feedback and constructive comments. Please find below the response to your concerns:**
>
> ---
> **Question 1:** It is not clear why the method is named and formalized as "flow matching" because the actual implementation is based on EDM. The perturbation and the denoising model both follow the diffusion custom instead of flow matching. Please justify whether the method, especially the probability path is closer to flow matching or diffusion.
>
> **Response:** Thank you for highlighting this important point. We acknowledge that the name "Flow Matching" might suggest a closer alignment with traditional flow matching methodologies. Our approach is indeed inspired by flow matching, particularly in how the forward process is deterministic, as described by the ODE in Eq. 1. However, by utilizing the Probability Flow (PF) formulation from diffusion models, we can reparameterize the vector field in Eq. 1 as a “denoiser” function. This integration allows us to leverage advancements in diffusion training, enhancing the robustness and performance of our model. Specifically, we adopt principles and hyperparameters from the EDM framework (Karras et al., 2022), which have been shown to make training more stable and produce superior outcomes in image generation tasks. In summary, our method is fundamentally rooted in flow matching through its deterministic forward process and that’s what has guided our naming convention.
>
> [Karras et al’22] Karras et al., *Elucidating the Design Space of Diffusion-Based Generative Models*, 2022
>
> ---
> **Question 2:** In Algorithm 2 the inputs to the denoising model $D_\theta$ are $(x,t)$ but elsewhere $(x, \sigma)$. This may lead to confusion on how the adaptive noise scaling can affect sampling.
>
> **Response:** Thank you for pointing this out. We acknowledge this can be a point of confusion. In EDM, the denoising model is conditioned on the noise level $\sigma_t$, which serves as a proxy for time $t$. In our case there is linear correspondence between sigma and time given by $\sigma_t := (1-t) \sigma_z$ (see Proposition 1). Hence we can easily convert time to sigma and vice versa. To maintain consistency, we have updated the notation to use $(x_t, \sigma_t)$ as inputs to the denoising model, throughout the paper (including Algorithm 2).
>
> ---
> **Question 3:** Application of the proposed method to other domains is unexplored, limiting the generalizability of the findings. It seems that the method can be applied to general image-to-image translation tasks, for example restoration of natural images which also involves stochasticity in generation of local details.
>
> **Response:** Thank you for your insightful suggestion. We agree that our framework can be directly applied to capturing fine details (with uncertainty) for image-to-image translation, and natural image restoration tasks. We have now highlighted this as promising future work in the conclusion section. However, our current study focuses on scenarios with significant “misalignment” in pixels and channels by leveraging the multiscale dynamics such having both deterministic and stochastic features. These challenges are more common in multiscale physics applications.
>
> ---
> **Question 4:** As acknowledged in the paper, SFM relies on paired input-output data, which may not always be feasible in real-world applications, particularly for unpaired or missing datasets.
>
> **Response:** Good observation. We acknowledge that this work relies on paired data, but this is the first step to solve the more challenging scenario with unpaired data. This can be considered as training data misalignment, that is another type of misalignment that is a natural extension of this study and we leave it for future research. We update the conclusions sections accordingly. Having said that, recent advancements in optimal-transport-based flow matching shows promising solutions to handle unpaired data in our case.
>
> ---
> **Question 5:** Improvement upon baseline method CorrDiff is not consistent (especially temperature in Table 2).
>
> **Response:** Great point. Temperature is inherently a rather deterministic variable. As a result, CorrDiff's two-stage approach (particularly the training of a large (deterministic) UNet regression in the first stage) effectively captures its behavior. Having said that, AFM consistently outperforms CorrDiff for rather stochastic channels and consistently offers better calibration of the spread for the ensemble members across **ALL** channels (measured by SSR).

---

> ### Author Response · Authors · 2024-11-26
> **Continued Authors Response to Reviewers Comments**
>
> Please find below the response to the rest of your concerns:
>
> **Question 6:** A minor point that does not affect the score (if the method is still modeled as flow matching). SFM/SfM is commonly used in the broad field of AI to refer to Structure from Motion, which could lead to confusion, particularly if the paper is targeted at a broader audience that includes computer vision researchers.
>
> **Response:** Thanks for pointing this out. We decided to change the name to “Adaptive Flow Matching” (AFM) to avoid confusion.
>
> ---
> **Question 7:** I wonder how the noise variance in the adaptive noise scaling varies in the training run, and if the effectiveness is sensitive to the hyperparameter setup.
>
> **Response:** Great point. We ran new experiments and added a new figure (see Fig. 7 in section B.2 of the revised manuscript) that shows the evolution of $\sigma_{\text{max}}$ over training steps for various channels (aka variables). Note that $\sigma_z$ can independently be tuned per channel. In a nutshell, $\sigma_z$ values initially increase across all channels due to high encoder error but stabilize as training progresses and the encoder's performance improves. Notably, the radar reflectivity channel maintains higher $\sigma_z$ values, reflecting its stochastic nature, while the temperature channel exhibits lower sigma values, aligning with its deterministic characteristics

---

> > ### Author Response · Authors · 2024-12-02
> >
> > Dear Reviewer Bz1Z,
> > Since the deadline for reviewers posting for the authors is coming soon, It would be great if you could acknowledge our response and let us know if further clarifications on any of your questions is required. Thanks for your help in the review process.
> > Best,
> > Authors

---

> > ### Comment · Reviewer_Bz1Z · 2024-12-02
> >
> > Thanks for the response that has addressed most of my concerns.
> >
> > Regarding the data-limited regime (although you mentioned that it is not the main concern of the paper), is it possible to include an ablation experiment on the size of the dataset to show the advantage of the proposed AFM in the limited data scenario, and whether the adaptiveness design or any hyperparameter choice can affect the performance under different scale of data availability? Sorry for my late question if the time is not enough to run the experiments. Anyway I would love to know your thoughts.

---

> > > ### Author Response · Authors · 2024-12-03
> > >
> > > We are glad that we have addressed most of your concerns.
> > >
> > > We totally agreed with the reviewer’s intuition about the importance of adaptive noise scaling for small data regime, and including an ablation on dataset size would be valuable to assess the merits of AFM. This ablation needs a full set of training and tuning that seems not doable within this short timeframe. We will however consider adding that to the final manuscript if the paper gets accepted.
> > >
> > > Having said that, we have added a new section B.1 to the appendix of the revised manuscript that includes further evidence about the encoder overfitting (see e.g., Fig. 7). It is obvious that overfitting happens quickly which is expected to be more severe when the size of the dataset becomes smaller. As a result, it’s natural to expect that the encoder sigma_z gets quickly inflated (during training) and thus leaves the heavy lifting for the generative flow matching to handle the uncertainty. Therefore, we expect the adaptive mechanism to be very helpful in smaller datasets when the overfitting is more severe.  Note also that adjusting the \lambda parameter to lower values could further prevent encoder overfitting.

---

> ### Author Response · Authors · 2024-12-03
>
> Dear reviewer Bz1Z,
>
> As the review period ends we would like to summarize our main updates in the revised manuscripts to address your concerns:
>
> - **Clarifying the synergy between AFM and EDM**: We make a clearer distinction between the FM and diffusion approaches. We have updated the entire paper to consistently use $(x_t, \sigma_t)$ as inputs to the denoising model and added further comments to clarify how we use EDM to reparameterize the AFM objective.
>
> - **Clarified Distinction from CorrDiff**: We elaborated on the key differences between our method and CorrDiff. We highlighted how our joint training approach with the $\lambda$ coefficient and adaptive noise scaling addresses overfitting and effectively handles data containing channels with varied stochasticity.
>
> - **Encoder Overfitting**: Added a new section B.1. with a set of new experiments (see e.g., Fig. 7, Tab. 5) showing how the UNet encoder overfits the data in a two-stage training regime and the impact in performance and ensemble variance.
>
> - **Analysis of adaptive noise scaling during training**: We added Section B.2 (incl. Fig. 8) in the revised manuscript that discusses the evolution of  \(\sigma_z\) over training steps for various channels (i.e. variables).
>
> Please let us know if further clarifications on any of your questions is required. Thanks for your help in the review process.
>
> Best,
> Authors

---

> > ### Comment · Reviewer_Bz1Z · 2024-12-03
> >
> > Thanks for the new content.
> >
> > About Fig. 7,
> > - I think it shows a generalisation gap between training and validation error curves, and at some point in the training stage the generalisation gap becomes more significant. However I don't see a clear sign of overfitting happening there such that the validation raises alongside training loss dropping.  In my view there might be some over-claiming. It is just generalization error rather than overfitting.
> > - also could you show the evidence of training and validation curves that your proposed method significantly reduces the generalisation gap (or tackles overfitting, if you would still claim that)?
> > - about your claim that overfitting "is expected to be more severe when the size of the dataset becomes smaller", could you show the evidence?

---

> > > ### Author Response · Authors · 2024-12-04
> > >
> > > Thanks for the feedback and constructive comments. Please find below the response to your concerns:
> > >
> > > **Question:** *About Fig. 7, I think it shows a generalisation gap between training and validation error curves, and at some point in the training stage the generalisation gap becomes more significant. However, I don't see a clear sign of overfitting happening there such that the validation raises alongside training loss dropping. In my view, there might be some over-claiming. It is just a generalization error rather than overfitting.*
> > >
> > > **Response:** Thank you for pointing this out. You're correct that the validation error doesn't increase significantly. However, we observed that the generalization gap widens as training progresses, which can be an early sign of overfitting. More importantly, longer training of the UNet negatively affects CorrDiff's ensemble variance across all variables and also leads to overfitting of the radar variable (see Table 5). This suggests that CorrDiff struggles to extrapolate beyond the training data. We acknowledge that our original wording might not have been clear. If our paper gets accepted, we'll make sure to clarify that our aim is to address overfitting and calibration for the task as a whole, not just for the UNet in isolation.
> > >
> > >
> > > **Question:** *Also, could you show the evidence of training and validation curves that your proposed method significantly reduces the generalisation gap (or tackles overfitting, if you would still claim that)?*
> > >
> > > **Response:** Thank you for the suggestion. For the adaptive $\sigma_z$ models, we logged both training and validation losses of the encoder. These show that the generalization gap remains relatively stable during training. Interestingly, even though the 1x1 convolution has a larger validation error, its generalization gap is smaller than the UNet's, which could in part explain its better results. We will include these training and validation curves in the final version of the paper if it gets accepted.
> > >
> > > **Question:** *About your claim that overfitting "is expected to be more severe when the size of the dataset becomes smaller," could you show the evidence?*
> > >
> > > **Response:** Thank you for bringing this up. We'll revise the statement to "is expected to be at least as pronounced when the size of the dataset becomes smaller," to better reflect the well-known findings in generalization theory.

---

### Author Response · Authors · 2024-12-03
**Summary of Revisions**

Dear Reviewers and Area Chairs,

As the review period concludes, we would like to thank you for your thoughtful and constructive feedback. We have carefully considered all your comments and have revised the manuscript accordingly. Below, we summarize the key changes and how we have addressed your main concerns:

- **Clarified Distinction from CorrDiff**: We elaborated on the key differences between our method and CorrDiff. We highlighted how our joint training approach with the $\lambda$ coefficient and adaptive noise scaling addresses overfitting and effectively handles data containing channels with varied stochasticity.

- **Evidence of Overfitting in CorrDiff**: Added new experimental evidence in Section B.1 (including Fig. 7 and Table 5) demonstrating overfitting issues of the UNet encoder and how it greatly affects ensemble variance in CorrDiff.

- **Improved Analysis of Adaptive Noise Scaling**: Added clarifications to Sections 4.2 and 4.3 and experimental results in Section B.2 discussing the evolution of $\sigma_z$ over training steps for various channels, providing insights into how the adaptive noise scaling operates and showcases how it handles the different stochasticity between variables.

- **Discussion on Adaptive $\sigma_z$**: While we opted for a simpler, closed-form solution for $\sigma_z$, we acknowledged that learning $\sigma$ through the encoder (e.g., using VAE approaches) is a viable alternative and added a remark discussing this potential extension.

- **Proof of Proposition 1**: We fixed the sign error in the proof of Proposition 1 and added further clarifications to ensure the clarity of the derivation. Provided a clearer definition of the velocity field in Proposition 1, showing how the $(1 - t)$ factor is accounted for in our formulation.

- **Explanation of Spatial Misalignment**: Clarified in the introduction how misaligned PDEs can result in spatially misaligned data due to divergent simulations, providing better context for our approach.

- **Related Work Discussion**: Acknowledged co-training methods in other domains (e.g., text-to-speech synthesis, precipitation nowcasting) and discussed them in Section 2. We emphasized how our work differs in task focus, handling of spatial misalignment, and multi-channel stochasticity.

- **Method Name**: Renamed our approach to "Adaptive Flow Matching" (AFM) to avoid confusion with established concepts.

- **Clarification between AFM and EDM**: Updated the entire paper to consistently use $(\mathbf{x}_t, \sigma_t)$ as inputs to the denoising model and added further comments to clarify how we use the Elucidated Diffusion Model (EDM) framework to reparameterize the AFM objective.

- **Hyperparameter Selection Details**: Provided more information on how we tune the $\lambda$ parameter using cross-validation to optimize RMSE on the validation set.

- **Identified Limitations and Future Work**: Added remarks in the conclusions section, noting that while our method's performance decreases with higher input-output misalignment (as shown in the Kolmogorov flow experiments), it still outperforms baseline models. We also recognized the importance of a theoretical convergence analysis of AFM and plan to pursue this in future work.

- **Corrections of Typos and Minor Errors**: Corrected all typos and minor errors pointed out, such as missing parentheses and repeated variables, to improve the overall readability and accuracy of the manuscript.

We hope that these revisions adequately address your concerns and improve the clarity and quality of our paper. We are grateful for your valuable feedback, which has helped strengthen our work.

Please let us know if there are any further questions or if additional clarifications are required.

Best regards,

The Authors

---

### Meta-Review · Area_Chair_vGmx · 2024-12-17

**Metareview:**

The paper introduces an Adaptive Flow Matching (AFM) framework for super-resolving small-scale physics, but reviewers raised concerns about its novelty and positioning relative to existing methods like CorrDiff and diffusion-based approaches. A key weakness highlighted is the lack of experiments in data-limited regimes, despite claims that the method is tailored for such settings, as well as insufficient evidence to demonstrate significant overfitting in baseline methods. Additionally, reviewers pointed out that the adaptive noise scaling mechanism and its theoretical justification require further clarity, particularly regarding its potential to degenerate into a fully stochastic model.

**Additional Comments On Reviewer Discussion:**

During the rebuttal period, reviewers raised concerns about the novelty of the method compared to existing approaches like CorrDiff, the lack of experiments in data-limited regimes, and the need for clearer justification of the adaptive noise scaling mechanism to prevent model degeneration. The authors addressed these by clarifying distinctions from CorrDiff, providing new evidence of overfitting in CorrDiff (e.g., training-validation divergence in Fig. 7), and explaining how adaptive noise scaling dynamically balances deterministic and stochastic components. However, I agree with reviewers that doubts remained regarding the lack of experiments in data-limited regimes which is one setting that AFM is tailored to.

---

### Decision · Program_Chairs · 2025-01-22

Reject